# The impact of companies disclosing ESG reports in multiple languages on the enthusiasm of foreign investors for holding shares

Ruixue Bao[1]*, Li Wei[2]

1 School of Accounting, Southwestern University of Finance and Economics, Chengdu, Sichuan, China,
2 School of Accounting, Guangxi Financial Vocational College, Nanning, Guangxi, China

* 2537218738@qq.com

**Data Availability Statement:** All data used in the study are publicly available from sources cited in the text. Interested persons can contact CSMAR for the data (see https://data.csmar.com for more

## Abstract

We collect Chinese A-share listed companies from 2013 to 2022 as samples and use the multi-period difference-in-difference model (DID) to study the impact of multilingual ESG report disclosure on the enthusiasm of foreign investors. We find that Chinese companies disclose ESG reports in both Chinese and English stimulate the enthusiasm of foreign investors to hold shares. The main manifestations are the expansion of the company's foreign shareholding quota and the increase in the number of shareholders. Further research show that disclosure of multilingual ESG reports makes up for the readability of company annual reports for foreign investors. In the case of companies with poor analyst attention and comparability of accounting information, and companies that hire non-big four auditing firms to audit financial reports, multilingual ESG report disclosures are more positive for foreign shareholdings. The participation of the central investor service center in corporate governance is weak, the degree of regional cultural integration is low, and the disclosure of English ESG reports by Chinese enterprises is conducive to promoting the enthusiasm of foreign shareholding. The research conclusions provide theoretical guidance and empirical reference for enterprises to expand information disclosure methods to foreign investors and attract overseas capital investment.

## 1.Introduction

In April 2023, the English website of the State Council reported Spanish investors' suggestions on information disclosure by Chinese listed companies. The Spanish investor suggested that companies listed on the Shanghai Stock Exchange and Shenzhen Stock Exchange publish financial performance reports in English like companies listed in Europe and Hong Kong, which would help foreign investors better understand the company's operations. Accounting and financial language is the universal language of business internationalization. The different languages in which companies disclose accounting and financial information will affect

details, contact via 400-639-888), CNRDS (see https://www.cnrds.com/ for more details, contact via 021-66181082), Official website of Shenzhen Stock Exchange (see https://www.szse.cn/ for more details, contact via 400-808-9999) and Official website of Shanghai Stock Exchange (see https://www.ssse.cn/ for more details, contact via 400-8888-400). We confirm that we do not have any special access or privileges to the data that other researchers would not have.

**Funding:** The author(s) received no specific funding for this work.

**Competing interests:** The authors have declared that no competing interests exist.

investors' understanding of the company's internal operations and the readability of listed company annual report information, which will affect investors' scientific decision-making [1,2]. Sustainable economic development is a topic of global concern, and the fulfillment of corporate environmental, social responsibility and corporate governance (ESG) responsibilities is an important factor affecting sustainable development. China's rapid economic growth is largely due to the large amount of overseas capital. Foreign investors are increasingly paying attention to ESG investments of Chinese companies. ESG reports are the mainstream information for global companies to disclose their fulfillment of sustainable development responsibilities, and will also tend to become a common language for international business communication.

As the world's largest developing country, China has contributed Chinese wisdom to promote the sustainable development of human society and actively promoted the sustainable development strategy. In 2018, the "Guidelines for the Governance of Listed Companies" promulgated by the China Securities Regulatory Commission forced that companies listed on the Shenzhen and Shanghai Stock Exchanges publish ESG reports on an annual basis. In addition to the company's annual financial report, ESG reports have become another annual report of non-financial information for companies. The company's annual financial report has multi-faceted interpretations by auditors, analysts and financial media, which helps provide a more convenient and efficient investment channel for foreign investors to participate in China's capital market investment. Compared with annual financial reports, ESG report content and format specifications have not yet been standardized, making it often difficult for ordinary investors without professional knowledge background to effectively identify and interpret text information. The language used in ESG reports is mainly Chinese, and most listed companies only choose the Chinese version to disclose ESG reports. ESG reports are an important carrier for corporate non-financial information disclosure. The diversification of Chinese context expressions affects the interpretation of text meanings. The large amount of Chinese text information provides greater space for management to manipulate language information and the readability of reports. The ESG report assurance of Chinese listed companies follows a voluntary principle. of China from 2006 to 2022, we have calculated that the proportion of companies conducting ESG report assurance is less than 2% (The results are shown in S1 Appendix). Third-party professional international rating agencies evaluate corporate ESG performance. Each ESG rating agency has large differences in underlying data sources, evaluation methods, and evaluation systems, resulting in significantly different rating results. In addition to referring to third-party professional international rating agencies such as Bloomberg for corporate ESG performance evaluations, there is little English version of ESG report information that investors can view directly on the company website. The international common language barrier and the imperfection of corporate ESG information quality assurance mechanisms have made it more difficult for foreign investors to understand ESG information, which has brought great confusion to investment decisions.

In 2023, International Sustainability Standards Board(ISSB) publishes IFRS S1 and IFRS S2 - "General Requirements for Disclosure of Sustainability-related Financial Information" and "Climate-related Disclosures". It will become a globally consistent standard for sustainable information disclosure and promote the internationalization of the ESG investment and financing system. According to the United Nations' World Investment Report 2021 (https://unctad.org/), China has become the world's largest foreign direct investment outflow country, with a total investment of $133 billion. In 2020, statistics from the United Nations Conference on Trade and Development (UNCTAD) show that China, which has ranked second for several consecutive years, surpassed the United States in the ranking of attracting foreign direct investment (FDI). The "Measures for the Administration of Information Disclosure by Listed

Companies" promulgated by the China Securities Regulatory Commission stipulates that listed companies can publish materials in English and other languages on their own according to the needs of international development. If Chinese companies want to develop sustainably, they must pay attention to their own ESG performance. Besides, how to balance internationalization and present ESG as a "business card" to gain recognition from international investors is the key to solving the problem.

We chose China scenario research for the following three reasons: first, China is the largest developing country in the world, and the fulfillment of ESG responsibilities by Chinese companies has an important influence on promoting a community with a shared future for human society. China's concept of harmonious social development places greater emphasis on the balanced integration of environment, society and corporate governance. In terms of the environment, the government advocates that "lucid waters and lush mountains are invaluable assets". In the social aspect, corporate social responsibility is emphasized, and the government promotes common prosperity and maintains social security. In terms of governance, government policies guide companies to emphasize both economic benefits and the protection of employee rights. The ultimate goal is to achieve synergy between enterprise and national development. Second, as China is the world's largest economy and investment country, the international versatility of ESG report disclosure languages affects investment decisions in the international capital market. In 2020 and 2021, China has become the world's largest country in terms of investment inflows and outflows, and Chinese companies need a wider range of foreign investors' capital participation. The State-owned Assets Supervision and Administration Commission of China has established a Social Responsibility Bureau to guide companies to proactively adapt to the formulation of international rules and standards, practice ESG theory, and make the languages chosen for corporate ESG reports more internationally applicable, further promoting the information transparency of Chinese companies to foreign investors, and stimulating the interest of foreign investors in holding shares. Third, China's reform and opening up adheres to the "bringing in" and "going out" strategies. The development of Chinese enterprises cannot be separated from the support of overseas capital. China's economic market coordinates the main role of the domestic macro-circulation and the mutual promotion of domestic and international dual cycles. The introduction of overseas capital into Chinese enterprises promotes the smooth circulation of production factors on a larger scale and promotes high-quality development of the world economy.

We use a multi-period difference-in-difference model (DID) to study the impact of multilingual ESG report disclosure on the enthusiasm of foreign investors based on a quasi-natural experiment in which Chinese companies disclose English versions of their ESG reports. The study found that Chinese companies choosing to disclose ESG reports in both Chinese and English can help stimulate the enthusiasm of foreign investors to hold shares. The main manifestation is that companies have expanded the scale of foreign shareholdings and increased the number of shareholders. Further research found that multilingual ESG report disclosure enhances the readability of company annual reports for foreign investors. The employment of the Big Four international accounting firms by companies to audit financial reports, analyst attention and comparability of accounting information affect the incentive effect of multilingual ESG report disclosure on foreign investors.

There are three innovations in this paper. First, it expands the research perspective on ESG report disclosure texts. Existing literature divides the disclosure structure of social responsibility reports into macro structure and micro structure, and studies the impact of the content framework structure of social responsibility reports on corporate disclosure of environmental responsibility [3]. The tone of environmental information disclosure reflects the performance of a company's environmental responsibilities. If a company's environmental responsibilities

are poorly fulfilled, an optimistic tone will be chosen for environmental information disclosure [4]. We selected the language type of ESG report disclosure text to conduct research to promote the ESG information disclosure of Chinese companies to adapt to international ESG standards.

Second, it is to explore the impact mechanism of ESG report disclosure language on cross-border capital flows. According to research literature from a micro perspective, corporate ownership structure, financial information disclosure quality and management agency problems are factors affecting foreign investor shareholdings in Chinese companies [5–7]. Regional market capacity, infrastructure and preferential policies have a positive effect on foreign investment, while wage costs have a negative effect on foreign investment [8]. Contract risks and political risks affect the scale of overseas capital investment [9]. From the perspective of non-financial information, we study the impact of ESG report disclosure languages on foreign capital investment in Chinese companies, and further enhance foreign investors' recognition of China's sustainable economic development.

Third, it enriches the research literature on corporate ownership structure. One type of literature is the study of the effects of the introduction of different types of equity on corporate governance and corporate performance [10]. Another type of literature is the study of factors affecting changes in corporate equity size [11]. Research on the introduction of cross-border capital by enterprises focuses on the effectiveness of equity governance effects [12], and few scholars focus on the advantageous strategies of Chinese enterprises in attracting cross-border capital [13]. The trend of sustainable development concepts has prompted cross-border investors to pay attention to corporate ESG reports, which provides us with research opportunities.

## 2.Literature review

### 2.1 Research on the impact of ESG information disclosure on corporate performance

The idiosyncratic information of enterprises in terms of strategic operations, social interests, and sustainable development is transmitted to investors, which affects the pricing efficiency of the capital market. ESG performance has an information effect. Enterprises with green disclosure requirements often have better market performance [14]. Better-performing companies disclose their sustainability performance in a way that is optimistic, certain, clear and more readable [15]. The higher ESG ratings contribute to a diminished likelihood of stock price crash risk, while the salutary impact is diminished by the prevalent ESG disagreement among rating agencies [16]. Larger companies tend to invest in ESG activities to better reflect the needs of their stakeholders. Companies with better media coverage can reduce information asymmetry among stakeholders in ESG investments [17]. ESG performance and ESG rating events with high financial investment behavior can promote corporate innovation [15]. ESG rating significantly improves stock liquidity [18]. Firms with sustainability requirements tend to have better market performance [19]. Companies reduce agency costs and improve corporate performance by fulfilling social responsibilities [15]. Companies with good ESG performance often have a strong sense of social responsibility, environmental awareness, and a high-level governance mechanism, which helps companies establish a high-quality image in the capital market [20]. Good ESG ratings and performance not only reduce information risks and financing costs, but also improve investment efficiency [21]. Low ESG ratings and negative news increase corporate operating risks, damage corporate value, and even endanger customer reputations, causing customer stock prices to fall [22]. Companies with good ESG performance integrate social responsibility into product differentiation strategies, build brand effects by

actively fulfilling social responsibilities to improve reputation, and produce differentiated high-quality products that are different from competitors [23].

Companies with high quality ESG information disclosure perform better, and debt financing costs play a partial mediating effect in the impact of ESG information disclosure on corporate performance [24]. Investors will evaluate a company's ESG behavior when investing and associate higher stock returns with the company's ESG management. The extent to which a company fulfills its ESG responsibilities represents the company's financial stability [25]. Corporate social responsibility information disclosure has a positive contribution to corporate financial performance and investment efficiency [26]. There is a positive relationship between CSR disclosure and the future financial performance of banks [19].

## 2.2 Research on the impact of text information characteristics on market investment

The textual characteristics of corporate non-financial information are important content that market investors pay attention to and are helpful for market investors' decision-making. The disclosure of data resources in corporate annual reports helps improve the efficiency of resource allocation in the capital market. The similarity between the management discussion and analysis (MD&A) text in the current annual report and the previous MD&A text indicates that the effectiveness of the company's information disclosure is weak and the risk of stock price crash may occur [26]. Annual report readability affects the persistence of corporate earnings, indicating management's opportunistic behavior [27]. The readability of annual reports is positively related to the company's investment efficiency [28], affects the accuracy and disagreement of analysts' forecasts [29], which causes an increase in the degree of disagreement in corporate bond ratings and debt financing costs rising [30]. Foreign auditors' interpretation of the Public Company Accounting Oversight Board (PCAOB) affects the comparability of international financial reports [31]. We set out to experimentally study the impact of report readability and the choice of managers' native or foreign language as the reporting language on risk taking in a management accounting context. The use of foreign languages in company annual reports combined with high readability can enhance investors' willingness to take beneficial risks [32].

## 2.3 Research on the impact of other information disclosure on corporate equity capital

Corporate private information, less public information creates undiversifiable risks for uninformed investors [33]. The disclosure of risk information helps investors understand the past, present and future risks faced by the company and narrows the gap between public and private information [34]. The two major indicators of accounting information transparency and accounting information comparability are negatively related to equity financing costs, and the impact of accounting information conservatism on equity financing costs is positively and significantly related. A high level of earnings quality translates into lower capital costs [35]. For companies with more serious accounting information asymmetry, their management may have concealed bad information, and the shareholding ratio of senior executives will be significantly reduced after listing. Companies with high transparency of information disclosure and low reliability of financial reports can help attract institutional investors to hold shares.

Disclosure of ESG information by companies can help promote and develop ESG responsible investment, encourage more institutional investors to participate in corporate governance, and thereby dilute equity and play a supervisory and restrictive role on major shareholders. In order to avoid poor stock liquidity caused by companies not disclosing information in a timely

manner, investors choose companies that disclose more information [36]. Companies with high quality information disclosure can reduce investors' expected risks in corporate investment, which promotes stock flow and reduce the cost of equity capital [37,38]. There is a negative relationship between a company's level of voluntary disclosure and the cost of equity capital [39]. Corporate voluntary carbon information disclosure and the release of social responsibility reports improve corporate information transparency and reduce the company's equity capital cost [40]. The quality of environmental information disclosure positively affects institutional investor shareholdings. Companies with high levels of annual report disclosure have lower equity financing costs [41]. There is a significant negative correlation between the quality of annual report credit information disclosure and the cost of equity capital. Better readability of board reports helps lower a company's cost of equity capital.

The impact of ESG information disclosure quality, text information characteristics and other information on market investors has been studied, but the impact of the language used in ESG information disclosure on overseas capital market investors has not yet been discussed. Considering the impact of cultural and language differences among foreign investors from mature markets, there may also be barriers to communication between foreign investors and local companies, and there may not be an effective way to obtain "soft information" from local companies [42], may rely more on public information channels [43]. The use of multiple languages by companies to disclose ESG reports can benefit all market participants and understand corporate sustainability responsibilities, rather than individual investors. This study is conducive to a deeper understanding of the governance value of capital market opening.

## 3.Theoretical analysis and research hypotheses

Enterprises' active performance of social responsibilities helps maintain and improve the relationship between enterprises and stakeholders. The stocks of enterprises with lower social responsibility performance can obtain excess returns in the short-term market, but corporate performance is unsustainable in the long-term market [44,45]. ESG reports carry non-financial information for companies to fulfill their social responsibilities. Text information is an explanation and supplement to digital information such as traditional financial statements, and has more information increments. ESG reports and financial reports have become the main basis for shareholders, creditors and other stakeholders to judge the sustainable development capabilities of enterprises. The ESG report contains a large amount of textual information. The textual content is forward-looking and predictive to a certain extent. Stakeholders can form expectations for the future development of the company based on the disclosed social responsibility information, which can help investors better understand the company's value and future prospects [46]. Currently, there are no international standards that uniformly regulate the disclosure content, form and language of corporate ESG reports. Companies can independently decide the content and form of ESG reports, which brings certain difficulties to investors in interpreting and analyzing ESG information. China's Shenzhen Stock Exchange and Shanghai Stock Exchange do not force listed companies to disclose relevant information reports in English. Most companies only disclose information reports in Chinese. If a company only discloses a Chinese version of its report, for foreign investors, communication language barriers will lead to a lack of readability of the ESG report. The readability of corporate annual reports directly affects the efficiency with which investors obtain information [47], and affects the accuracy of information that stakeholders trust [48].

First, the English version of ESG reports disclosed by companies affects the efficiency of foreign investors in obtaining information. Foreign investors' participation in the Chinese market will help promote the effective flow and allocation of resources and increase stock liquidity

[36]. The "Measures for the Administration of Information Disclosure by Listed Companies" promulgated by the China Securities Regulatory Commission stipulates that listed companies can publish materials in English and other languages on their own according to the needs of international development. As an annual report of corporate non-financial information, the ESG report discloses the English version of the ESG report. It is an international communication method and becomes a communication bridge between Chinese companies and foreign-owned companies. The ESG report is an annual report of corporate non-financial information. Disclosure of the English version of the ESG report is an international communication method and has become a bridge of communication between Chinese companies and foreign-owned shares. The English version of the ESG report can reduce the communication costs and information processing costs between foreign investors and Chinese enterprises, and solve the decision-making problems of overseas investors more conveniently and quickly. The English version of the ESG report improves the information environment of the capital market, broadens the efficiency of information use in the international market, helps improve the decision-making quality of information users in the capital market, and increases the willingness of foreign investors to trade [2].

Second, the English version of ESG reports disclosed by companies affects the accuracy of the information that foreign investors trust. Corporate social responsibility is a kind of "on-the-job consumption" of management. Management often over-invests in social responsibility-building activities to enhance their professional and personal reputation. Social responsibility disclosure may further strengthen management's tendency to violate regulations [49]. CSR disclosures provide management with a "legitimate justification" and a "license" to engage in financial irregularities [50]. Therefore, the communication costs between foreign investors and local corporate managers and other shareholders are usually higher, making the information asymmetry between foreign investors and insiders more serious. Due to the profit-seeking nature of capital, the future performance and earnings of enterprises are the focus of investors' decision-making. Compared with local Chinese investors, it is more difficult for foreign investors to grasp and predict the company's future development and profitability. Companies disclosing ESG reports in English can help enhance the readability of corporate ESG report information for foreign investors. The easier it is for foreign investors to interpret the contents of ESG reports of Chinese companies, the more accurately they can judge the company's operating conditions and investment risks [41]. Choosing English to disclose information in ESG reports maximizes the familiarity of foreign shareholders, enhances the impact of corporate internationalization, and promotes foreign shareholders' recognition of corporate culture and identity. Foreign investors have reduced their doubts about the operation and management of Chinese companies, increased their trust in companies' public disclosure of information, and increased investors' willingness to invest [51].

Hypothesis 1: Multilingual ESG report disclosure is significantly positively related to the enthusiasm of foreign investors for holding shares.

## 4. Research design

### 4.1 Source and selection of research samples

In 2018, the China Securities Regulatory Commission revised the "Code of Governance for Listed Companies" and added the "Stakeholders, Environmental Protection and Social Responsibility" chapter, establishing for the first time the basic framework for environmental, social responsibility and corporate governance (ESG) information disclosure. Article 95

stipulates that "listed companies shall disclose environmental information and performance of social responsibilities such as poverty alleviation in accordance with laws, regulations and the requirements of relevant departments." Article 96 stipulates that "listed companies shall disclose corporate governance-related information in accordance with relevant regulations, regularly analyze the status of corporate governance, formulate plans and measures to improve corporate governance, and implement them conscientiously." Starting in 2018, the China Securities Regulatory Commission has mandated that companies disclose social responsibility reports or ESG reports. This paper uses 2018 as the natural experiment time to conduct research. In order to control the influence of bias in the two periods before and after the implementation of the policy on the robustness of the test results, we need to maintain the balance of the research time interval of the experimental group and the treatment group. We selected samples five years before the experimental event and five years after the experimental event for research. This paper selects Chinese A-share listed companies from 2013 to 2022 as a sample to study the impact of multilingual ESG report disclosure on foreign shareholdings. The research samples of this paper come from the official website of Shenzhen Stock Exchange, the official website of Shanghai Stock Exchange, China Stock Market & Accounting Research Database(This database is referred to as CSMAR) and Chinese Research Data Services Database(This database is referred to as CNRDS). Considering the impact of special samples on the robustness of research conclusions, the following research samples were eliminated: (1) Enterprise samples in the financial industry; (2) ST and *ST enterprise samples; (3) Enterprise samples that did not disclose social responsibility reports; (4) Samples with partial missing data. Foreign shareholders have investment preferences, which creates a sample self-selection problem for foreign-invested enterprises. ST indicates that the company's net income has been negative in the past two years, that is, the listed company has suffered losses for two consecutive years or its net assets are lower than the face value of the stock. Or the company's shareholders' rights in the last year were lower than its registered capital. *ST means that the company has suffered losses for three consecutive years and is at risk of being withdrawn from the stock market. This paper uses the propensity score matching method (This method is referred to as PSM) proposed by Rosenbaum and Rubin (1983) to control the endogeneity problem. We winnow all study variables at the 1% level.

Existing literature research has concluded that corporate governance level, enterprise age, debt level, enterprise nature, growth level, etc. are important factors affecting foreign investment choices [52,53]. This paper selects enterprise size (Size), asset profit rate (Roa), debt ratio (*Lev*), long-term debt ratio (*Lon*), and the largest shareholder's shareholding ratio (*Top1*) as the parameters variables, companies that disclose multilingual ESG reports are used as the experimental group, and control group samples are selected according to the 1:4 proximity matching method. We describe the sample screening process in detail in Table 1.

## 4.2 Definition of research variables

China vigorously promotes the construction of pilot free trade zones and free trade ports, actively promotes exchanges with international capital markets, and improves the degree of liberalization and facilitation for Chinese overseas investors, with a view to attracting a large amount of foreign investment into China's capital market. The high-level opening up of China's capital market has further strengthened the confidence of global investors to participate and enhanced the enthusiasm of foreign investors for holding shares. China Securities Daily reported that there are two aspects that highlight the enthusiasm of foreign investors holding shares in Chinese listed companies. First, the scale of foreign investors purchasing shares in China's A-share market has increased. Second, the number of foreign investors has increased.

**Table 1. Sample screening.**

| Panel A Treat Group | |
| --- | --- |
| **Sample classification** | **Number** |
| Sample of companies disclosing ESG reports in multiple languages | 155 |
| Exclude the financial industry sample | 42 |
| Remaining samples | 113 |
| **Panel B Control Group** | |
| **Sample classification** | **Number** |
| Sample of companies disclosing ESG reports in multiple languages | 11788 |
| Exclude the financial industry sample | 1841 |
| Exclude ST and ST* enterprise samples | 2167 |
| Eliminate samples of companies that do not support PSM(the propensity score matching method) | 7328 |
| Remaining samples | 452 |

We use the scale of foreign shareholding and the number of foreign shareholders to measure the enthusiasm of foreign investors to hold shares in Chinese listed companies.

**4.2.1 Scale of foreign shareholding.** The scale of foreign shareholding refers to the shareholding ratio of Chinese overseas legal persons and natural persons in listed companies in China, and the sum of the shareholding ratios of foreign shareholders among the top ten shareholders of Chinese listed companies. Considering the endogeneity problem of reverse causation between explanatory variables and explained variables, this paper uses the increase in foreign shareholding ratio in the current period compared with the foreign shareholding ratio in the previous period to measure the scale of foreign ownership (*ΔSharehold*).

**4.2.2 Number of foreign shareholders.** Shanghai Securities News commented,"Foreign capital has become one of the most important investor categories in China's A-share market. The continued increase in the number of investors shows that overseas investors are more cognizant of China's medium and long-term development prospects and are relatively optimistic about the prospects of China's high-quality development." This paper uses the natural logarithm of the number of foreign equity holders in listed companies in China to measure the number of foreign shareholders (*Shareholdnum*).

**4.2.3 Multilingual ESG report disclosure.** Before 2018, the disclosure of sustainable development information by listed companies in China regarding the fulfillment of social responsibilities and other sustainable development information was voluntary. In 2018, Articles 95 and 96 of China's "Code of Governance for Listed Companies" mandated companies to disclose environmental information, fulfill Situations related to social responsibility and corporate governance. This paper defines dummy variables based on the time node of the quasi-natural experiment study in 2018. Before the English version of the ESG report was disclosed, the variable Time is assigned a value of 0, and after the English version of the ESG report is disclosed, the variable *Time* is assigned a value of 1.

Most ESG reports disclosed by listed companies in China are mainly in Chinese, and some companies voluntarily choose to disclose ESG reports in both Chinese and English. This research defines companies that disclose ESG reports in both Chinese and English as the experimental group, with a value of *Treat* = 1, and companies that only use Chinese to disclose ESG reports as a control group, with a value of *Treat* = 0.

**4.2.4 Control variables.** Due to language and cultural barriers, there are differences between foreign shareholders, local corporate managers, and other shareholders. The information asymmetry between foreign shareholders and the company is serious, making it more

difficult for foreign shareholders to grasp and predict the company's future development and profitability. The future performance and income of a company affect the enthusiasm of foreign shareholders to invest in Chinese listed companies [10,28]. China has a vast territory, and differences in the level of market economic development in various regions affect the degree of attraction of the capital market to foreign investors [12]. The degree of diversification of corporate ownership structures affects the participation of major shareholders within the company in relation to foreign shareholders [10]. Enterprise ownership structure, financial performance characteristics and China's regional macro factors affect the enthusiasm of foreign capital to invest in Chinese listed companies.

This paper refers to the existing literature on capital structure and equity governance and controls the following relevant variables at the company level and regional level [7–9,28,34]. Company-level control variables: enterprise size (*Size*), the total assets at the end of the period are taken as the natural logarithm. The age of enterprise establishment (*Age*) is the logarithmic value of the current year of the sample period minus the year of establishment of the enterprise. Profit rate on assets (*Roa*), the ratio of operating profit to total assets. Debt ratio (*Lev*), the ratio of total liabilities at the end of the period to total assets at the end of the period. Long-term debt ratio (*Lon*), the ratio of long-term liabilities at the end of the period to total assets at the end of the period. The shareholding ratio of the largest shareholder (*Top1*) is the ratio of the number of shares held by the largest shareholder to the total number of shares. Nature of property rights (*Soe*), a dummy variable, is assigned a value of 1 for state-controlled enterprises, otherwise it is assigned a value of 0. Technology-intensive industries (*Techind*), dummy variable, technology-intensive industries are assigned a value of 1, otherwise they are assigned a value of 0.

Control variables at the regional level [12,42]: gross urban product (*LNGDP*), which is the logarithm of the actual gross product of the prefecture-level city where the enterprise is located. City employment level (*City_employ*), which is the logarithm of the total number of employees in the prefecture-level city where the enterprise is located. City wage level (*City_R&D*), the proportion of R&D investment in GDP in the prefecture-level city where the enterprise is located. Regional variable (*Area*): based on the geographical and economic connection characteristics of each region in our country, we divide the country into six major regions and set corresponding dummy variables 1–6 respectively. Among them, the northeastern region includes Liaoning, Jilin and Heilongjiang, the Bohai Rim region includes Beijing, Tianjin, Hebei and Shandong, the southeastern region includes Shanghai, Jiangsu, Zhejiang, Fujian and Guangdong, the central region includes Henan, Hubei, Hunan, Anhui and Jiangxi, and the southwest The region includes Chongqing, Sichuan, Yunnan, Guizhou, Guangxi and Hainan, and the northwest region includes Shanxi, Shaanxi, Gansu, Ningxia, Inner Mongolia, Xinjiang, Qinghai and Tibet. Time variable (*Year*): from 2013 to 2022, set 10 annual dummy variables. Individual variable (*Firm*): company dummy variable. We define these variables in Table 2.

## 4.3 Model construction

After the company's ESG report is disclosed, investors can further strengthen their understanding of the company's non-financial information through the ESG report and reduce the degree of information asymmetry of the company. After the disclosure of multilingual ESG reports, English as the common language is more convenient for foreign investors to understand the ESG information of enterprises, which affects the convenience of foreign investors in making shareholding decisions of domestic enterprises. In 2018, the China Securities Regulatory Commission revised the "Guidelines for Governance of Listed Companies" to establish a

**Table 2. Research variable definitions.**

| Variables | Definition |
|---|---|
| $\Delta$Sharehold | The increase in foreign shareholding ratio in this period compared with the foreign shareholding ratio in the previous period. |
| Shareholdnum | The number of foreign equity holders in listed companies in China is calculated as the natural logarithm. |
| Time | Before the company discloses the English version of the ESG report, the variable Time is assigned a value of 0. After the company discloses the English version of the ESG report, the variable Time is assigned a value of 1. |
| Treat | Companies that use both Chinese and English to disclose ESG reports are the experimental group, and are assigned Treat = 1. Companies that only use Chinese to disclose ESG reports are the control group, and are assigned Treat = 0. |
| Size | Take the natural logarithm of total assets at the end of the period. |
| Age | The logarithmic value of the current year of the sample period minus the year of establishment of the company. |
| Roa | Operating profit to total assets ratio |
| Lev | Ratio of total liabilities at the end of the period to total assets at the end of the period. |
| Lon | Ratio of long-term liabilities at the end of the period to total assets at the end of the period. |
| Top1 | The proportion of the number of shares held by the largest shareholder to the total number of shares. |
| Soe | State-controlled enterprises are assigned a value of 1, otherwise they are assigned a value of 0. |
| LnGDP | The logarithm of the actual GDP of the prefecture-level city where the enterprise is located. |
| City_employ | The logarithm of the total number of employees in the prefecture-level city where the enterprise is located. |
| City_R&D | Proportion of R&D investment in GDP of the prefecture-level city where the enterprise is located. |
| Area | According to the geographical and economic connection characteristics of each region in our country, we divided the country into six major regions and set corresponding dummy variables 1–6 respectively. |
| Techind | Technology-intensive industries are assigned a value of 1, otherwise they are assigned a value of 0. |
| Year | From 2013 to 2022, set 10 annual dummy variables. |
| Firm | Firm individual dummy variable. |

corporate ESG information disclosure framework for the first time, mandating companies to start disclosing social responsibility reports or ESG reports. The establishment of a system for Chinese listed companies to disclose ESG reports has formed a policy basis for companies to disclose ESG reports in multiple languages. This policy basis is an exogenous impact event for companies. In the years after 2018, companies gradually disclosed ESG reports in both Chinese and English. Evaluating policy effects under traditional methods mainly conducts regression tests by setting dummy variables of whether companies disclose ESG reports in English. In comparison, our research sample is panel data, and our research method using the difference-in-differences model model is more scientific, which avoids the problem of reverse causation and more accurately estimate the policy effects of companies using multiple languages to disclose ESG reports.

Starting in 2018, companies gradually began to disclose English versions of ESG reports. This paper constructed a multi-period difference-in-differences model (DID) to test the rationality of the research hypotheses. In order to solve the endogeneity problem caused by the mutual causal relationship between variables, we conducted the following processing when constructing a multi-period difference-in-difference model (DID). In the difference-in-differences model (1) we set up, the explained variable is the difference between the foreign shareholding ratio in the current period and the foreign shareholding ratio in the previous period,

which represents the increase in the shareholding scale of foreign investors. In the difference-in-differences model (2) we set up, the explained variable is the number of foreign shareholders of the enterprise in the current period, and the explanatory variables and control variables are lagged by one period. At the same time, the fixed effects of region, technology-intensive industry, year and individual company level are controlled, which also alleviates the problem of omitted variable bias to a certain extent.

$$
\begin{aligned}
\Delta \text{Sharehold}_{i,t} ={}& \beta_0 + \beta_1 \text{Time}_{i,t} \times \text{Treat}_{i,t} + \beta_2 \text{Size}_{i,t} + \beta_3 \text{Age}_{i,t} + \beta_4 \text{Roa}_{i,t} \\
& + \beta_5 \text{Leverage}_{i,t} + \beta_6 \text{Lev}_{i,t} + \beta_7 \text{Top1}_{i,t} + \beta_8 \text{Soe}_{i,t} + \beta_9 \text{LnGDP}_{j,t} \\
& + \beta_{10} \text{City\_employ}_{j,t} + \beta_{11} \text{City\_R\&D}_{j,t} + \gamma_i + \eta_i + \lambda_t + \mu_i + \varepsilon_{i,t}
\end{aligned} \tag{1}
$$

$$
\begin{aligned}
\text{Sharenum}_{i,t+1} ={}& \beta_0 + \beta_1 \text{Time}_{i,t} \times \text{Treat}_{i,t} + \beta_2 \text{Size}_{i,t} + \beta_3 \text{Age}_{i,t} + \beta_4 \text{Roa}_{i,t} \\
& + \beta_5 \text{Leverage}_{i,t} + \beta_6 \text{Lev}_{i,t} + \beta_7 \text{Top1}_{i,t} + \beta_8 \text{Soe}_{i,t} + \beta_9 \text{LnGDP}_{j,t} \\
& + \beta_{10} \text{City\_employ}_{j,t} + \beta_{11} \text{City\_R\&D}_{j,t} + \gamma_i + \eta_i + \lambda_t + \mu_i + \varepsilon_{i,t}
\end{aligned} \tag{2}
$$

In the difference-in-differences model (1) and (2), *Time* is a dummy variable. Before the English version of the ESG report is disclosed, the variable *Time* is assigned a value of 0. After the English version of the ESG report is disclosed, the variable *Time* is assigned a value of 1. Before 2018, the sustainable development information of listed companies in China regarding the fulfillment of social responsibilities and other the disclosure is voluntary. Articles 95 and 96 of China's 2018 "Code of Governance for Listed Companies" mandate that companies disclose environmental information, fulfill social responsibilities and corporate governance. Starting in 2018, it is officially confirmed that corporate ESG reports can be disclosed in multiple languages. The experimental group represents companies that disclose ESG reports in both Chinese and English, and is assigned *Treat* = 1. The control group represents companies that only disclose ESG reports in Chinese, and is assigned Treat = 0. *Size*, *Age*, *Roa*, *Lev*, *Lon*, *Top1*, *Soe*, *LnGDP*, *City_employ*, *and City_R&D* represent the control variables. $\gamma_i$, $\eta_i$, $\lambda_t$ and $\mu_i$ represent the fixed effect of the control area, the fixed effect of the technology-intensive industry, the time fixed effect, and the individual enterprise effect, respectively. $\varepsilon_{i,t}$ is the random error term of the model, in order to further avoid the impact of unobservable variables on the empirical results of the paper. The variable we are interested in is the interaction term *Time×Treat*, which represents that companies' use of multiple languages to disclose ESG reports increases the enthusiasm of foreign investors. We expect $\beta_1$ to be positive. The $\beta_2 \sim \beta_{11}$ represent the regression coefficients corresponding to the control variables.

## 5. Descriptive statistics and empirical results analysis

### 5.1 Descriptive statistics

Table 3 shows the descriptive statistics results of the sample. The average value of the variable *Shareholdnum* is 2.0232, the 75% quantile is 2.0000, and the maximum value is 8.0000, indicating that the average number of foreign shareholders in Chinese enterprises is small. The average value of variable *ΔSharehold* is -0.6893, the median is 0.0000, the 75% quantile is 0.2700, and the maximum value is 32.8000, indicating that more than 50% of foreign investors have a positive growth rate of shareholding size, and more than 50% of overseas investments The growth rate of investors' shareholding size is negative, but the growth level of foreign investors' shareholding size is low. The average value of variable *Treat* is 0.2741, the average value of variable *Post* is 0.2568, and the 75% quantile is 1.0000, indicating that the proportion of Chinese capital market companies disclosing ESG reports in multiple languages is relatively low.

**Table 3. Descriptive statistical results of research sample.**

| VARIABLES | N | mean | sd | min | P25 | P50 | P75 | max |
|---|---|---|---|---|---|---|---|---|
| *Shareholdnum* | 565 | 2.0232 | 1.5322 | 1.0000 | 1.0000 | 1.0000 | 2.0000 | 8.0000 |
| *ΔSharehold* | 565 | -0.6893 | 6.1752 | -28.6100 | -0.5300 | 0.0000 | 0.2700 | 32.8000 |
| *Treat* | 565 | 0.2741 | 0.4465 | 0.0000 | 0.0000 | 0.0000 | 1.0000 | 1.0000 |
| *Post* | 565 | 0.2568 | 0.4373 | 0.0000 | 0.0000 | 0.0000 | 1.0000 | 1.0000 |
| *Size* | 565 | 23.5269 | 1.7051 | 20.2687 | 22.2151 | 23.4479 | 24.7835 | 26.6145 |
| *Age* | 565 | 2.9161 | 0.3462 | 1.3863 | 2.7081 | 3.0445 | 3.1355 | 3.5553 |
| *Roa* | 565 | 0.1071 | 0.0905 | -0.2737 | 0.0582 | 0.1098 | 0.1560 | 0.3517 |
| *Lev* | 565 | 0.4562 | 0.1908 | 0.0446 | 0.3119 | 0.4642 | 0.6059 | 0.8602 |
| *Lon* | 565 | 0.0577 | 0.0847 | 0.0000 | 0.0000 | 0.0169 | 0.0848 | 0.3612 |
| *Top1* | 565 | 37.3774 | 16.8465 | 2.4307 | 24.5559 | 3.5693 | 49.6837 | 79.7343 |
| *Soe* | 565 | 0.3598 | 0.4804 | 0.0000 | 0.0000 | 0.0000 | 1.0000 | 1.0000 |
| *LnGDP* | 565 | 6.5251 | 3.8542 | -6.6709 | 6.5800 | 6.6000 | 7.0109 | 13.6200 |
| *City_employ* | 565 | 96.2617 | 0.5945 | 95.8500 | 95.9400 | 95.9600 | 96.0200 | 98.6000 |
| *City_R&D* | 565 | 3.4835 | 0.8697 | 0.5100 | 3.5600 | 3.6600 | 3.7300 | 6.4400 |

## 5.2 Analysis of main regression empirical results

In Table 4, the intersection multiplication term *Treat×Post* in column (1) has a significant positive correlation with the variable *Shareholdmun* at the 1% level. The intersection multiplication term *Treat×Post* in column (2) has a significant positive correlation with the variable *ΔSharehold* at the 1% level. This indicates that the multi-lingual disclosure of ESG reports by companies improves foreign investors' understanding of Chinese companies' ESG responsibility fulfillment, alleviates the information asymmetry between foreign investors and Chinese companies, and increases the enthusiasm of foreign capital holders to invest in Chinese companies. The research hypothesis 1 is verified.

## 5.3 Robustness test

**5.3.1 The Placebo test.** The impact of corporate multilingual disclosure of ESG reports on green total factor productivity may be interfered by random factors or other policies. In order to identify whether changes in the enthusiasm for foreign ownership of the treatment group and the control group are affected by random factors or other policies, this paper conducts a placebo test two years in advance of the year the policy starts to be implemented. In Table 5, the p value corresponding to the DID regression coefficient of the virtual treatment group and the virtual policy time interaction term is not significant at the 10% level, indicating that the placebo test results support the robustness of the baseline regression results.

**5.3.2 Parallel trend testing and policy dynamic effect analysis.** The parallel trend assumption is the premise for using the DID model to evaluate policy effects. It assumes that the target variables of the treatment group and the control group have the same change trend before the policy occurs. The dynamic effect test is to determine whether the impact of multilingual disclosure of ESG reports on the enthusiasm of foreign-owned Chinese companies changes over time, whether it has a continuous driving effect or shows a certain degree of lag.

**Table 4. Basic regression results.**

| VARIABLES | (1) | (2) |
|---|---|---|
| | *Shareholdmun* | *ΔSharehold* |
| *Treat×Post* | 4.9061*** | 25.3692*** |
| | (4.7377) | (5.6842) |
| *Size* | -0.5378 | -1.1931 |
| | (-0.9326) | (-1.2457) |
| *Age* | 1.3267 | -18.8751** |
| | (0.5048) | (-2.1142) |
| *Roa* | -0.6058 | 2.2598 |
| | (-0.4128) | (0.9962) |
| *Lev* | 3.6931* | 7.3045 |
| | (1.7952) | (1.4423) |
| *Lon* | 2.0746 | -4.0660 |
| | (0.7303) | (-0.5523) |
| *Top1* | -0.0345 | 0.0906 |
| | (-1.0672) | (0.6630) |
| *Soe* | -3.3709** | -5.7507** |
| | (-2.0473) | (-2.1798) |
| *LnGDP* | 0.1648*** | 0.4289*** |
| | (3.6852) | (3.7340) |
| *City_employ* | 4.6016** | 32.6201*** |
| | (2.5160) | (5.0724) |
| *City_RD* | 0.9771 | 41.4981*** |
| | (0.1789) | (3.9648) |
| *Area* | Yes | Yes |
| *Techind* | Yes | Yes |
| *Year* | Yes | Yes |
| *Firm* | Yes | Yes |
| *Constant* | -4.3459** | -3.1880*** |
| | (-2.1740) | (-5.0621) |
| Observations | 565 | 565 |
| R-squared | 0.1368 | 0.2841 |

Note:We report t-statistics based on robust standard errors clustered by firm in parentheses, while ***, **, * denote statistical significance at 1%, 5%, and 10%levels, respectively.

In order to test the parallel trend assumption and analyze the dynamic effects of policies, the following econometric model is constructed based on model (1) and model (2):

$$
\begin{aligned}
\Delta \text{Sharehold}_{i,t} &= \beta_0 + \beta_1 \sum \textit{Bef}_{i,s} + \beta_2 \sum \text{Cur}_{i,t} + \beta_3 \sum \text{Aft}_{i,m} + \beta_4 \text{Size}_{i,t} + \beta_5 \text{Age}_{i,t} + \beta_6 \text{Roa}_{i,t} \\
&+ \beta_7 \text{Leverage}_{i,t} + \beta_8 \text{Lev}_{i,t} + \beta_9 \text{Top1}_{i,t} + \beta_{10} \text{Soe}_{i,t} + \beta_{11} \text{LnGDP}_{j,t} \\
&+ \beta_{12} \text{City\_employ}_{j,t} + \beta_{13} \text{City\_R\&D}_{j,t} + \gamma_i + \eta_i + \lambda_t + \mu_i + \varepsilon_{i,t}
\end{aligned} \tag{3}
$$

**Table 5. The Placebo test results.**

| VARIABLES | (1) | (2) |
|---|---|---|
|  | *Shareholdmun* | *ΔSharehold* |
| *DID* | -0.1211 | -15.2445 |
|  | (-0.0270) | (-1.5411) |
| *Size* | -0.4958 | -0.9789 |
|  | (-0.8766) | (-0.9922) |
| *Age* | 1.5147 | -17.9101** |
|  | (0.5855) | (-1.9704) |
| *Roa* | -0.4330 | 3.1578 |
|  | (-0.3026) | (1.3902) |
| *Lev* | 3.7164* | 7.4187 |
|  | (1.8141) | (1.4771) |
| *Lon* | 1.9153 | -4.8921 |
|  | (0.6767) | (-0.6760) |
| *Top1* | -0.0350 | 0.0881 |
|  | (-1.0875) | (0.6630) |
| *Soe* | -3.2645** | -5.2031* |
|  | (-2.0147) | (-1.9261) |
| *LnGDP* | -0.0053 | -0.4523 |
|  | (-0.0592) | (-1.3215) |
| *City_employ* | -2.5532 | -4.3889 |
|  | (-0.9535) | (-0.7529) |
| *City_RD* | -0.5791 | 33.4564** |
|  | (-0.0972) | (2.2553) |
| *Area* | Yes | Yes |
| *Techind* | Yes | Yes |
| *Year* | Yes | Yes |
| *Firm* | Yes | Yes |
| *Constant* | 2.5861 | 3.9910 |
|  | (0.8945) | (0.6512) |
| Observations | 565 | 565 |
| R-squared | 0.1312 | 0.2563 |

Note: We report t-statistics based on robust standard errors clustered by firm in parentheses, while ***, **, * denote statistical significance at 1%, 5%, and 10%levels, respectively.

$$
\begin{aligned}
\text{Sharenum}_{i,t+1} &= \beta_0 + \beta_1 \sum \text{Bef}_{i,s} + \beta_2 \sum \text{Cur}_{i,t} + \beta_3 \sum \text{Aft}_{i,m} + \beta_4 \text{Size}_{i,t} + \beta_5 \text{Age}_{i,t} + \beta_6 \text{Roa}_{i,t} \\
&\quad + \beta_7 \text{Leverage}_{i,t} + \beta_8 \text{Lev}_{i,t} + \beta_9 \text{Top1}_{i,t} + \beta_{10} \text{Soe}_{i,t} + \beta_{11} \text{LnGDP}_{j,t} \\
&\quad + \beta_{12} \text{City\_employ}_{j,t} + \beta_{13} \text{City\_R\&D}_{j,t} + \gamma_i + \eta_i + \lambda_t + \mu_i + \varepsilon_{i,t}
\end{aligned}
\tag{4}
$$

In the difference-in-differences model (3) and (4), the variables $Bef_{i,s}$, $Cur_{i,t}$ and $Aft_{i,m}$ respectively represent the cross-multiplication term of the year dummy variable and the corresponding experimental group dummy variable s years before the policy implementation, the t year of the current policy implementation period, and m years after the policy implementation. $\beta_1$, $\beta_2$, and $\beta_3$ represent the regression coefficients of variables $Bef_{i,s}$, $Cur_{i,t}$, and $Aft_{i,m}$

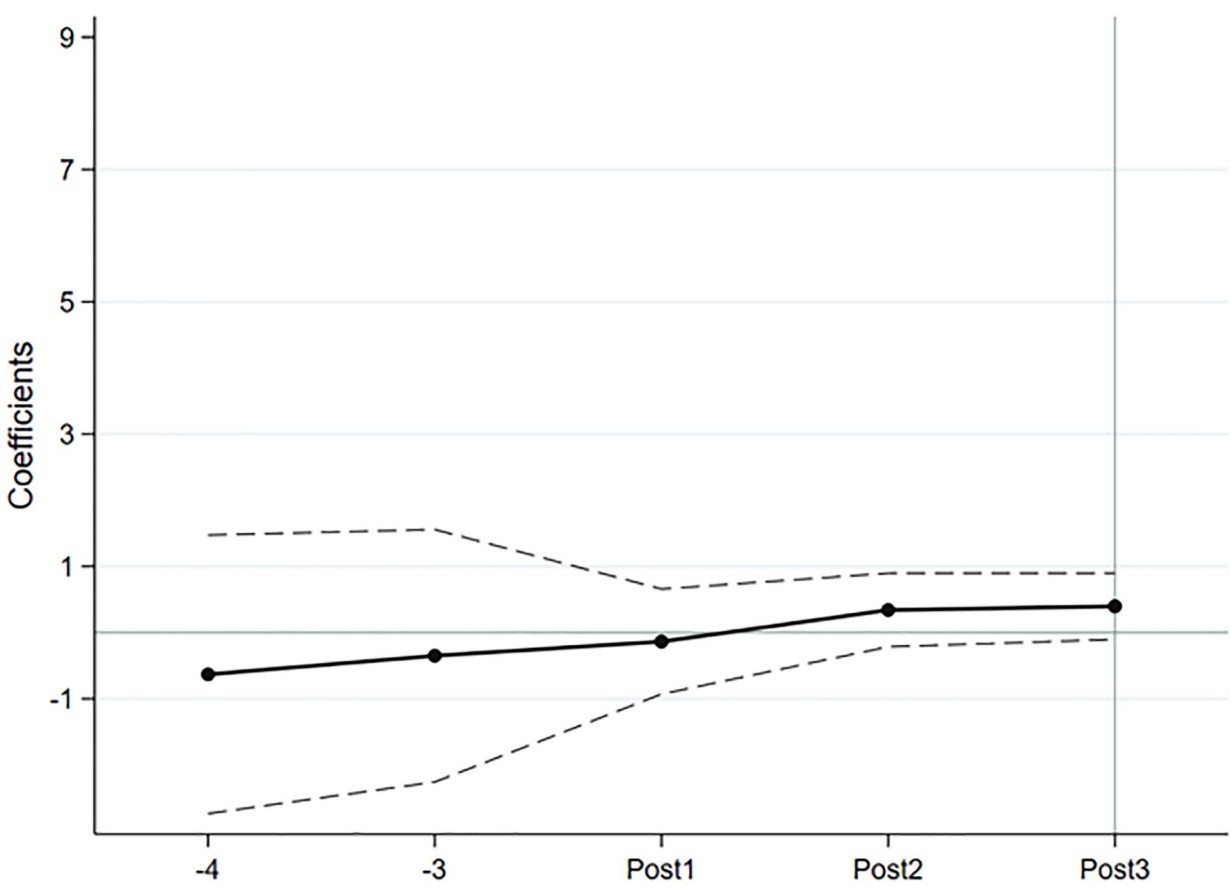

**Fig 1. Parallel trend test in the number of foreign investors and analysis of policy dynamic.**

respectively. *Size*, *Age*, *Roa*, *Lev*, *Lon*, *Top1*, *Soe*, *LnGDP*, *City_employ*, and *City_R&D* represent the control variables. The $\beta_4 \sim \beta_{13}$ represent the regression coefficients corresponding to the control variables. $\gamma_i$, $\eta_i$, $\lambda_t$ and $\mu_i$ represent the fixed effect of the control area, the fixed effect of the technology-intensive industry, the time fixed effect, and the individual enterprise effect, respectively. $\varepsilon_{i,t}$ is the random error term of the model.

The results of the parallel trend test and policy dynamic effect analysis are shown in Figs 1 and 2. The confidence interval of the coefficient of the interaction term from the early stage of policy implementation all intersects with the 0 axis, indicating that the treatment group and the control group have the same trend of change before the implementation of the policy, which satisfies the assumption of parallel trends. Overall, the regression coefficient increases year by year, indicating that multilingual disclosure of ESG reports has a dynamic effect, and its impact on the number of foreign investors and the size of foreign shareholdings is increasing.

**5.3.3 Exclude the impact of the pilot policy on paid use and trading of energy rights.**
To respond to climate change and promote green and low-carbon development, companies need to bring into play the role of market mechanisms. Beginning in 2016, four provinces, Zhejiang, Fujian, Henan, and Sichuan, issued the "Pilot Implementation Plan for Paid Use and Trading of Energy Rights". Since the policy on paid use and trading of energy rights overlaps with the ESG report disclosure policy, it is possible that multilingual ESG reports have less

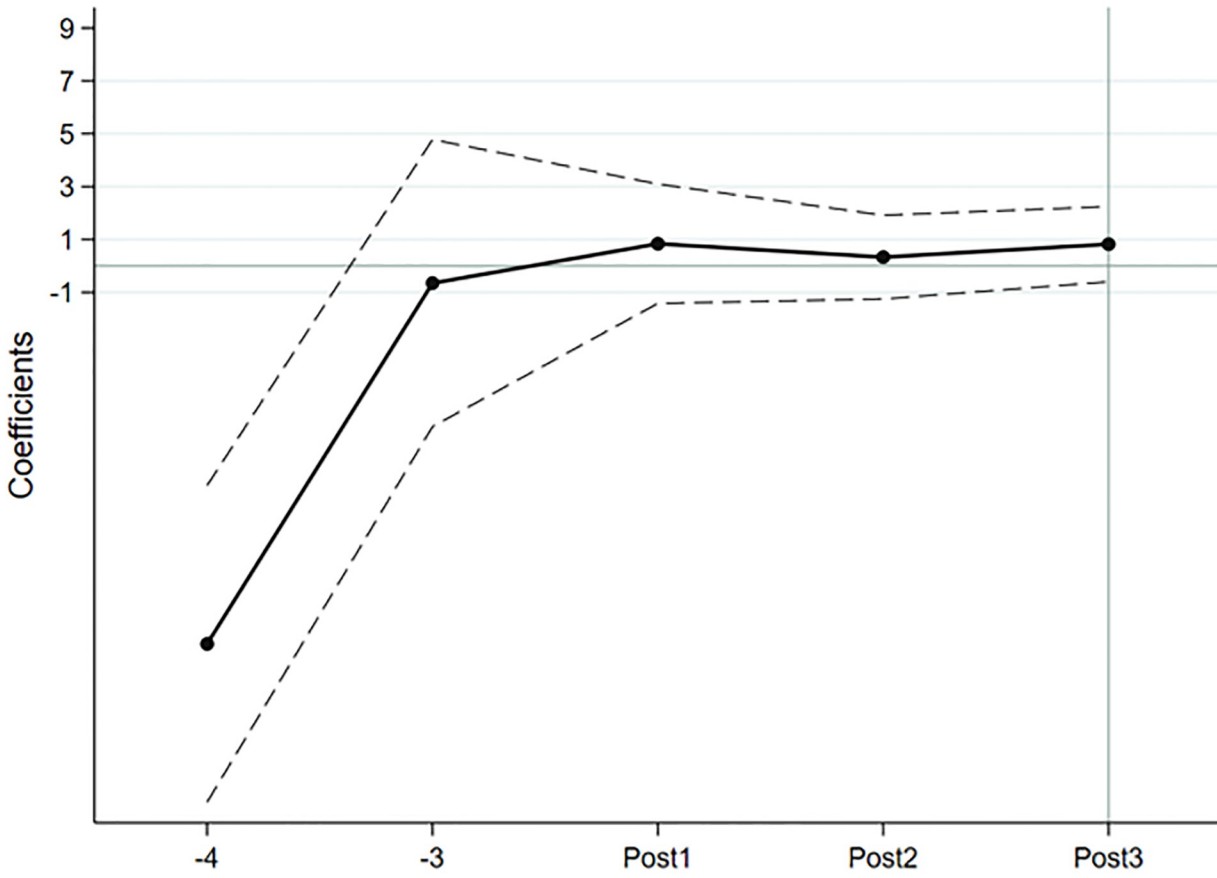

**Fig 2. Parallel trend test of foreign shareholding scale and analysis of policy dynamic effects.**

impact on the enthusiasm of foreign investors for holding shares. In order to avoid the interference of the paid use and trading policies of energy rights on the research results, this paper eliminates the above four provinces from the sample to further test the robustness of the benchmark regression results. The regression results are shown in Table 6.

## 6.Further research

The purpose of companies using multiple languages to disclose ESG reports is to meet the information needs of foreign investors and attract the enthusiasm of foreign investors to hold shares in domestic companies. The mechanism by which multilingual disclosure of ESG reports affects the enthusiasm of foreign investors, and the effect of multilingual disclosure of ESG reports needs further research.

### 6.1 Inspection of impact mechanism

Most Chinese domestic enterprises disclose ESG reports in Chinese for domestic capital market investors. To achieve the goal of world-class enterprises, Chinese enterprises need ESG concepts to be transmitted to the international capital market. The ESG information of Chinese enterprises is internationalized, and the use of English to disclose ESG reports can help enhance the efficiency of foreign investors in obtaining ESG information of Chinese enterprises, and strengthen foreign investors' trust in the accuracy of ESG information of Chinese

**Table 6. Robustness test results excluding special samples.**

| VARIABLES | (1) | (2) |
|---|---|---|
| | *Shareholdmun* | *ΔSharehold* |
| *Treat×Post* | 3.0061*** | 15.2291*** |
| | (2.7887) | (3.6935) |
| *Size* | -0.6135 | -1.2241 |
| | (-1.0639) | (-1.3898) |
| *Age* | -0.2474 | -18.6424*** |
| | (-0.0887) | (-2.7576) |
| *Roa* | -0.9732 | 1.9933 |
| | (-0.6641) | (0.6325) |
| *Lev* | 4.1531** | 7.8532** |
| | (2.0436) | (2.1098) |
| *Lon* | 1.6121 | -6.2287 |
| | (0.5438) | (-1.0181) |
| *Top1* | -0.0363 | 0.0918 |
| | (-1.0498) | (1.4368) |
| *Soe* | -3.9056** | -6.2957 |
| | (-2.5130) | (-1.6401) |
| *LnGDP* | 0.0035 | 0.0308 |
| | (0.0945) | (0.2174) |
| *City_employ* | -1.2421 | 0.6922 |
| | (-1.4095) | (0.2066) |
| *City_RD* | -2.3962 | 10.3891* |
| | (-0.8840) | (1.8667) |
| *Area* | Yes | Yes |
| *Techind* | Yes | Yes |
| *Year* | Yes | Yes |
| *Firm* | Yes | Yes |
| *Constant* | 1.464 | -1.0591 |
| | (1.6045) | (-0.0324) |
| Observations | 285 | 285 |
| R-squared | 0.1288 | 0.2747 |

Note: We report t-statistics based on robust standard errors clustered by firm in parentheses, while ***, **, * denote statistical significance at 1%, 5%, and 10%levels, respectively.

enterprises. Investment decision needs the reliability of financial information. The disclosure of ESG report is an important non-financial information report of an enterprise. The environmental responsibility, social responsibility and corporate governance concerned in ESG report can all be reflected in the financial information of an enterprise. ESG report disclosure and financial information become complementary functions, which can become the information basis for investors to understand the authenticity of financial information. Chinese enterprises' use of English to disclose ESG reports has improved the readability of corporate financial reports for foreign investors.

Chinese enterprises use English to disclose ESG reports to supplement the difficulty of foreign investors in interpreting Chinese enterprises' financial reports and enhance the readability of foreign investors' financial reports. The improvement of the readability of the text helps

to enhance the efficiency of foreign investors in obtaining information. By using the ESG reporting information of Chinese enterprises, foreign investors can understand the financial reporting information of Chinese enterprises, understand the implementation of ESG responsibilities of enterprises, and further trust the authenticity of financial information. Low readability of annual report text information will reduce investors' investment in enterprises [51]. Multilingual disclosure of ESG reports by enterprises can promote the readability and understanding of financial information by foreign investors, which is an influence mechanism to stimulate the enthusiasm of foreign shareholding.

This paper uses the annual report readability measurement index from WenGou Finance Database, and takes a negative number from the calculation formula of WenGou Finance Database to indicate poor readability of the annual report. The greater the Readability value, the worse the readability of the text. This paper constructs the following regression model according to the three-step method [54], and performs regression to test the mechanism by which multilingual disclosure of ESG reports affects the enthusiasm of foreign investors.

$$
\begin{aligned}
\text{Readability}_{i,t} &= \beta_0 + \beta_1 \text{Time}_{i,t} \times \text{Treat}_{i,t} + \beta_2 \text{Size}_{i,t} + \beta_3 \text{Age}_{i,t} + \beta_4 \text{Roa}_{i,t} \\
&\quad + \beta_5 \text{Leverage}_{i,t} + \beta_6 \text{Lev}_{i,t} + \beta_7 \text{Top1}_{i,t} + \beta_8 \text{Soe}_{i,t} + \beta_9 \text{LnGDP}_{j,t} \\
&\quad + \beta_{10} \text{City\_employ}_{j,t} + \beta_{11} \text{City\_R\&D}_{j,t} + \gamma_i + \eta_i + \lambda_t + \mu_i + \varepsilon_{i,t}
\end{aligned} \tag{5}
$$

$$
\begin{aligned}
\Delta \text{Sharehold}_{i,t} &= \beta_0 + \beta_1 \text{Time}_{i,t} \times \text{Treat}_{i,t} + \beta_2 \text{Readability}_{i,t} + \beta_3 \text{Size}_{i,t} + \beta_4 \text{Age}_{i,t} + \beta_5 \text{Roa}_{i,t} \\
&\quad + \beta_6 \text{Leverage}_{i,t} + \beta_7 \text{Lev}_{i,t} + \beta_8 \text{Top1}_{i,t} + \beta_9 \text{Soe}_{i,t} + \beta_{10} \text{LnGDP}_{j,t} \\
&\quad + \beta_{11} \text{City\_employ}_{j,t} + \beta_{12} \text{City\_R\&D}_{j,t} + \gamma_i + \eta_i + \lambda_t + \mu_i + \varepsilon_{i,t}
\end{aligned} \tag{6}
$$

$$
\begin{aligned}
\text{Shareholdnum}_{i,t+1} &= \beta_0 + \beta_1 \text{Time}_{i,t} \times \text{Treat}_{i,t} + \beta_2 \text{Readability}_{i,t} + \beta_3 \text{Size}_{i,t} + \beta_4 \text{Age}_{i,t} + \beta_5 \text{Roa}_{i,t} \\
&\quad + \beta_6 \text{Leverage}_{i,t} + \beta_7 \text{Lev}_{i,t} + \beta_8 \text{Top1}_{i,t} + \beta_9 \text{Soe}_{i,t} + \beta_{10} \text{LnGDP}_{j,t} \\
&\quad + \beta_{11} \text{City\_employ}_{j,t} + \beta_{12} \text{City\_R\&D}_{j,t} + \gamma_i + \eta_i + \lambda_t + \mu_i + \varepsilon_{i,t}
\end{aligned} \tag{7}
$$

Among them, *Time* is a dummy variable. Starting in 2018, it is officially confirmed that corporate ESG reports can be disclosed in multiple languages. Before the English version of the ESG report is disclosed, the variable *Time* is assigned a value of 0. After the English version of the ESG report is disclosed, the variable *Time* is assigned a value of 1. The experimental group represents companies that disclose ESG reports in both Chinese and English, and variable *Treat* is assigned a value of 1. The control group represents companies that only disclose ESG reports in Chinese, and variable *Treat* is assigned a value of 0. We expect that corporate disclosure of ESG reports in multiple languages will help improve the readability of the company's annual report, and the regression coefficient $\beta_1$ of the difference-in-difference model (5) will be negative. The variable *Readability* is added to the difference-in-differences model (6) and (7) for testing. We predict that the readability difficulty of the company's annual report will be significantly negatively related to the enthusiasm of foreign investors, and the regression coefficient $\beta_1$ will be negative.

*Size*, *Age*, *Roa*, *Lev*, *Lon*, *Top1*, *Soe*, *LnGDP*, *City_employ*, and *City_R&D* represent the control variables. In the difference-in-difference model (5), $\beta_2 \sim \beta_{11}$ represent the regression coefficients corresponding to the control variables. In the difference-in-differences model (6) and (7), $\beta_2$ represents the regression coefficient of the mediating variable *Readability*, and $\beta_3 \sim \beta_{11}$ represent the regression coefficients corresponding to the control variables. $\gamma_i$, $\eta_i$, $\lambda_t$ and $\mu_i$ represent the fixed effect of the control area, the fixed effect of the technology-intensive industry,

**Table 7. Influence mechanism test results.**

| VARIABLES | (1) | (2) | (3) |
|---|---|---|---|
| | *Readability* | *Shareholdmun* | *ΔSharehold* |
| Readability | — | -4.5942** | -15.6721*** |
| | — | (-2.1417) | (-3.2443) |
| Treat×Post | -0.0156** | -0.1089 | 1.6061 |
| | (-2.4609) | (-0.4266) | (0.3337) |
| Size | -0.0078*** | -0.0493 | -0.4709 |
| | (-8.0118) | (-1.2204) | (-0.5062) |
| Age | -0.0088** | 0.2464 | -14.4456*** |
| | (-2.2393) | (1.6449) | (-3.9261) |
| Roa | -0.0022 | 0.2468 | -2.5001 |
| | (-0.1552) | (0.4541) | (-0.2297) |
| Lev | -0.0050 | 1.0962*** | -5.7863 |
| | (-0.6213) | (3.5703) | (-0.8783) |
| Lon | 0.0035 | -0.7110 | 14.9445 |
| | (0.2236) | (-1.1615) | (1.1438) |
| Top1 | -4.53e-05 | -0.0015 | 0.3910*** |
| | (-0.5622) | (-0.4933) | (4.1626) |
| Soe | 0.0162*** | -0.0025 | -12.9891*** |
| | (5.0204) | (-0.0183) | (-4.8576) |
| LnGDP | -0.0006 | -0.0152 | -0.2876 |
| | (-1.3354) | (-0.7981) | (-0.9958) |
| City_employ | -0.0067** | 0.0052 | -4.5389 |
| | (-2.1048) | (0.0400) | (-1.5031) |
| City_RD | -0.0010 | 0.1687* | 3.2443** |
| | (-0.6116) | (1.7382) | (1.9813) |
| Area | Yes | Yes | Yes |
| Techind | Yes | Yes | Yes |
| Year | Yes | Yes | Yes |
| Firm | Yes | Yes | Yes |
| Constant | 0.6621** | -0.2412 | 0.4407 |
| | (2.1498) | (-0.0192) | (1.5254) |
| Observations | 565 | 565 | 565 |
| R-squared | 0.3656 | 0.1099 | 0.3021 |

Note: We report t-statistics based on robust standard errors clustered by firm in parentheses, while ***, **, * denote statistical significance at 1%, 5%, and 10%levels, respectively.

the time fixed effect, and the individual enterprise effect, respectively. $\varepsilon_{i,t}$ is the random error term of the model.

In column (1) of Table 7, the intersection multiplication term *Treat×Post* has a significant negative correlation with the variable *Readability* at the 5% level. The variable *Readability* in column (2) has a significant negative correlation with the variable *Shareholdmun* at the 5% level, and the variable *Readability* in column (3) has a significant negative correlation with the variable *ΔSharehold* at the 1% level. Disclosure of ESG reports in multiple languages by companies improve the readability of the company's annual report and make up for the understanding of the content of the company's annual report by foreign investors.

## 6.2 Additional inspection

**6.2.1 The impact of analyst attention.** Due to the existence of information asymmetry, investors use media tracking reports and analyst tracking reports to reduce the information asymmetry of their investments before investing. Analysts are important participants in listed companies' conference calls, performance briefings, and on-site surveys. Analysts provide incremental information to investors by observing the tone and intonation of management's responses to questions. Securities analysts can improve the accuracy of forecasts based on their financial accounting knowledge, industry research, etc.[55]. The research reports disclosed by analysts help to present the private information of the company to investors and facilitate investors to make a more accurate assessment of the value of the company. The intensity of analysts' follow-up attention will significantly improve the quality of information disclosure of listed companies [56]. Based on their professional expertise, analysts can better interpret the financial information of listed companies and allow listed company information to be quickly disseminated in the capital market by publishing research reports [57]. A larger number of analysts tracking means more intense competition among analysts, more effective monitoring function [58], and analysts are more likely to dig out and discover negative information about the company in the process of researching the company. A larger number of analysts following the company allows the company's negative information to spread faster and more widely, thus bringing greater operating pressure to corporate managers [59].

Companies with higher analyst attention have higher information transparency. Foreign investors can obtain interpretations of corporate financial and non-financial information through multiple channels. Foreign investors rely less on ESG reports for investment decisions of Chinese companies. Analysts pay less attention to companies, and investors have limited access to information resources. As a supplement to financial information, ESG reports have important reference value for investors. Disclosure of ESG reports in multiple languages increases the channels for foreign investors to understand non-financial information, alleviates confusion in understanding financial information, and enhances the information transparency of companies' overseas investments. Therefore, differences in analyst attention affect the attractiveness of multilingual ESG reports to foreign investors.

This paper uses the number of analyst followings to measure analyst attention. First, we sort the companies from small to large according to the number of analysts following the companies by industry and year. Second, we divided the sorted samples into three groups based on one-third quantiles and two-thirds quantiles. Third, we use the first one-third group of samples and the last one-third group of samples to conduct group regression tests to compare the differences in analysts' attention and the impact of multilingual disclosure of ESG reports on the enthusiasm of foreign investors.

In column (1) of Table 8, the intersection multiplication term *Treat×Post* has a significant positive correlation with the variable *Shareholdmun* at the 10% level. In column (2), there is no significant correlation between the intersection multiplication term *Treat×Post* and the variable *Shareholdmun*. Companies with low analyst attention, foreign investors lack professional interpretation of Chinese company annual reports, and multilingual ESG report information disclosures have strengthened the number of foreign investors. In Table 8, the intersection multiplication term *Treat×Post* in column (3) has a significant positive correlation with the variable *ΔSharehold* at the 10% level. In column (2), there is no significant correlation between the intersection multiplication term *Treat×Post* and the variable *ΔSharehold*. For companies with low attention from investors, disclosing ESG reports in multiple languages increase the enthusiasm of foreign-owned Chinese companies.

**Table 8. Impact results of differences in analyst attention.**

| VARIABLES | (1) | (2) | (3) | (4) |
|---|---|---|---|---|
| | Analyst attention is low | Analyst attention is high | Analyst attention is low | Analyst attention is high |
| | Shareholdmun | | ΔSharehold | |
| Treat×Post | 1.1589* | -0.0998 | 8.1671* | 3.3652 |
| | (1.8152) | (-0.2759) | (1.6837) | (0.4031) |
| Size | 0.0323 | -0.3574** | 0.1422 | -5.0443*** |
| | (0.1896) | (-2.2723) | (0.1181) | (-2.6715) |
| Age | 0.517 | 1.0193** | -3.5567 | -23.0011** |
| | (0.9749) | (2.0031) | (-0.5303) | (-2.0888) |
| Roa | 0.0454 | 2.7711* | 2.0678 | 39.3223 |
| | (0.0246) | (1.7612) | (0.1797) | (1.6290) |
| Lev | 1.367 | 2.6851** | 1.4921 | 12.3891 |
| | (1.4906) | (1.9612) | (0.1474) | (0.7601) |
| Lon | -1.5912 | 1.4191 | 11.5824 | 49.5960* |
| | (-0.6456) | (0.5300) | (0.4885) | (1.7975) |
| Top1 | -0.0131 | 0.0112 | 0.3383** | 0.4933** |
| | (-1.1473) | (1.0782) | (2.3339) | (2.3075) |
| Soe | 0.3512 | 0.9953* | -13.2901*** | -7.9945 |
| | (0.8197) | (1.9440) | (-3.5732) | (-0.9284) |
| LnGDP | -0.1352** | -0.0037 | -16.0121** | -0.4412 |
| | (-1.9658) | (-0.1651) | (-2.2655) | (-0.8530) |
| City_employ | -0.5435 | -0.3162 | 10.7566 | -4.3510 |
| | (-1.5216) | (-1.3683) | (0.8150) | (-1.0351) |
| City_RD | 0.0240 | 0.5489*** | 18.3782*** | 3.9108 |
| | (0.1873) | (3.3264) | (8.1672) | (0.3365) |
| Area | Yes | Yes | Yes | Yes |
| Techind | Yes | Yes | Yes | Yes |
| Year | Yes | Yes | Yes | Yes |
| Firm | Yes | Yes | Yes | Yes |
| Constant | 5.1452 | 3.4261 | -9.4442 | 5.6878 |
| | (1.4938) | (1.5085) | (-0.7848) | (1.3484) |
| Observations | 420 | 145 | 420 | 145 |
| R-squared | 0.0921 | 0.5043 | 0.2432 | 0.5335 |
| Coefficient difference test | Prob > chi2 = 0.0003 (Treat×Post) | | Prob > chi2 = 0.0000 (Treat×Post) | |

Note: We report t-statistics based on robust standard errors clustered by firm in parentheses, while ***, **, * denote statistical significance at 1%, 5%, and 10%levels, respectively.

**6.2.2 The impact of auditing by the big four international auditing firms.** The big four international auditing firms have a long history in the international capital market, their auditing business is spread all over the world. They have accumulated rich auditing professional skills, and they are highly recognized in the capital market. The audit business of the big four international auditing firms not only focuses on financial information, but also has the ability to conduct ESG report assurance business. In the process of financial auditing by the big four international auditing firms, ESG information is used to confirm the objective reliability of financial information. Most auditing firms in China have not yet implemented ESG report assurance services, and are unable to evaluate and judge financial information more accurately,

and the level of investor protection is low. Compared with companies audited by the big four international auditing firms, foreign investors have less credibility with company information audited by Chinese domestic auditing firms. For companies audited by domestic auditing firms in China, disclosing ESG reports in English helps foreign investors interpret financial information and increase the enthusiasm of foreign investors to hold shares.

If an enterprise hires a big four international auditing firm to audit the financial report, we define the value as 1 for the Big Four; otherwise, the value is 0. We divide the research sample into two groups based on whether the company employed a big four international auditing firm. We conduct group regression to examine the differences in audits by the four major international auditing firms, and the impact of multilingual disclosure of ESG reports on the enthusiasm of foreign investors.

In column (1) of Table 9, there is no significant correlation between the cross product *Treat×Post* and the variable *Shareholdmun*. In column (2), the intersection multiplication term *Treat×Post* is significantly positively correlated with the variable *Shareholdmun* at the 10% level. In column (3), there is no significant correlation between the intersection multiplication term *Treat×Post* and the variable *ΔSharehold*. In column (4), the intersection multiplication term *Treat×Post* has a significant positive correlation with the variable *ΔSharehold* at the 5% level. The company does not employ the big four auditing firms, and foreign investors cannot fully trust the annual report information of Chinese companies in a Chinese-language context. Multilingual ESG report information disclosure strengthens the trust level of foreign investors and attracts more foreign investors.

**6.2.3 The impact of comparability of accounting information.** The comparability of accounting information means that when different accounting entities have the same or similar economic business, they produce similar accounting information. When the economic business is different, they produce accounting information with certain differences [60]. Obtaining incremental information from the comparative analysis of accounting information comparability and competitors in the same industry can help investors obtain more information related to decision-making. The comparability of accounting information helps information users compare, analyze and identify the company's current accounting information with its own historical accounting information, giving the company's stakeholders a benchmark for evaluating managers' behavior. The comparability of accounting information can also help shareholders and managers sign and implement more efficient and reasonable compensation contracts, improve managers' performance-salary sensitivity, enhance the incentive effect of compensation contracts, and suppress the company's agency costs. When the comparability of corporate accounting information is poor, it is more difficult for foreign investors to use non-financial information reports in the Chinese context of Chinese companies to grasp true and reliable information within the company, which affects their enthusiasm for investment decisions in domestic companies. Companies disclose ESG reports in multiple languages to make up for the shortcomings of financial reporting information reporting. It is a company's detailed interpretation of the fulfillment of ESG responsibilities, which improves the readability of information for overseas investors. The transparency of corporate information in overseas capital markets strengthens investors' trust levels.

This paper draws on the measurement method of accounting information comparability [61], constructs model (8) and uses the company's data in the first 16 quarters of period t to estimate the company's accounting system. In the model, the explained variables $Earning_{i,t}$ (calculated as the company's quarterly net profit divided by the market value at the beginning of the period) represents financial reporting, and the explanatory variable $Return_{i,t}$ (expressed

**Table 9. The impact of the audit status of the four major international auditing firms.**

| VARIABLES | (1) | (2) | (3) | (4) |
|---|---|---|---|---|
| | Big Four | Non-Big Four | Big Four | Non-Big Four |
| | Shareholdmun | | ΔSharehoId | |
| Treat×Post | -2.7931 | 2.3278* | 12.1091 | 39.4950** |
| | (1.8707) | (1.6877) | (0.7785) | (2.5744) |
| Size | -0.1596 | -0.5121 | -0.7782 | 0.5161 |
| | (-0.8684) | (-0.8052) | (-0.2991) | (0.5040) |
| Age | 1.0345 | -1.2121 | -17.0012** | -10.7011** |
| | (1.2232) | (-0.5035) | (-2.0281) | (-2.4846) |
| Roa | 3.9462 | -1.4789 | 16.0123 | 6.8811 |
| | (1.6122) | (-0.9413) | (0.5262) | (0.5749) |
| Lev | 1.3608 | 5.7091** | 52.5221*** | -16.1182** |
| | (0.8332) | (2.5430) | (2.7077) | (-2.0914) |
| Lon | 7.1694* | 0.6741 | -1.2181** | 3.0242** |
| | (1.8756) | (0.2196) | (-2.3193) | (2.0898) |
| Top1 | -0.0013 | -0.0357 | 0.3789* | 0.5021*** |
| | (-0.1038) | (-1.0680) | (1.9186) | (4.3156) |
| Soe | 0.8164** | -4.9773*** | -24.7657*** | -13.5321*** |
| | (2.0779) | (-2.8965) | (-4.8298) | (-4.5253) |
| LnGDP | 0.0281 | 0.0024 | 3.3843** | -0.5910** |
| | (0.4716) | (0.0685) | (2.0405) | (-1.9890) |
| City_employ | -0.6876 | -1.3409 | 4.9731 | -3.0076 |
| | (-1.4981) | (-1.3010) | (0.4551) | (-0.7919) |
| City_RD | 0.3162 | -1.5213 | 8.2663 | 2.8956 |
| | (0.6161) | (-0.7576) | (0.7682) | (1.5963) |
| Area | Yes | Yes | Yes | Yes |
| Techind | Yes | Yes | Yes | Yes |
| Year | Yes | Yes | Yes | Yes |
| Firm | Yes | Yes | Yes | Yes |
| Constant | -1.5292 | 1.5443 | -4.6437 | 2.7968 |
| | (-0.0822) | (0.7553) | (-0.4388) | (0.7734) |
| Observations | 104 | 461 | 104 | 461 |
| R-squared | 0.6741 | 0.1822 | 0.4889 | 0.3069 |
| Coefficient difference test | Prob > chi2 = 0.0042 (Treat×Post) | | Prob > chi2 = 0.0000 (Treat×Post) | |

Note: We report t-statistics based on robust standard errors clustered by firm in parentheses, while ***, **, * denote statistical significance at 1%, 5%, and 10%levels, respectively.

as quarterly stock return) represents economic business.

$$\text{Earning}_{m,t} = \alpha_I + \beta_I \, \text{Return}_{i,t} + \varepsilon_{i,t} \tag{8}$$

Use formula (8) to calculate the expected earnings of company i in period t based on the average conversion function of the accounting system of industry I when the same economic business (Return$_{i,t}$ is the same).

$$E(\text{Earning})_{i,t} = \hat{\alpha}_I + \hat{\beta}_I \text{Return}_{i,t} \tag{9}$$

Formula (9), the coefficient $\hat{\alpha}_I$ and $\hat{\beta}_I$ represents the average accounting system conversion

function of industry I, E(Earning)i, t represents the expected earnings of company i in period t.

According to De Franco et al. (2011) definition [61], the more similar the accounting systems of two companies are, the more comparable the accounting information will be. Assume that the economic businesses of company i and company j are $Return_{i,t}$ and $Return_{j,t}$ respectively. The average accounting system conversion functions $\hat{\alpha_i}$ and $\hat{\beta_i}$ of industry I estimated by model (9) are substituted into model (10) and model (11) respectively, and the calculation is The expected earnings of company i and the expected earnings of company j.

$$E(Earning)_{i,t} = \hat{\alpha}_i + \hat{\beta}_i Return_{i,t} \tag{10}$$

$$E(Earning)_{i,t} = \hat{\alpha}_j + \hat{\beta}_j Return_{j,t} \tag{11}$$

The expected earnings calculated through the accounting systems of company i and company j are obtained from models (10) and (11) respectively. The difference between Company$_i$'s expected earnings and Company$_j$'s expected earnings indicates the comparability of the accounting information of the two companies. In order to make the results more robust, we take the average of the previous sixteen quarters as a proxy indicator of the comparability of accounting information. Finally, according to model (12), the accounting information comparability (*Comp Acct$_{i,j,t}$*) of company i and company j is obtained.

$$\text{Comp Acct}_{i,j,t} = \frac{1}{16} \times \sum_{t}^{t=16} |E(Earning)_{i,t} - E(Earning)_{j,t}| \tag{12}$$

In Eq (12), we refer to the method of the literature (De Franco et al., 2011), 16 means using each company's earnings and quarterly stock returns in the past 16 quarters for estimation. According to the above method, the absolute value of the difference between the expected accounting earnings of the two companies is calculated, and then the average of the absolute difference between the expected accounting earnings of the two companies is calculated in sixteen consecutive quarters, and the reverse number of the final calculation result is taken to define the accounting information comparable of company i and company j in the period t. Repeat the above steps to calculate the mean comparability of all paired combinations of company i in period t and other companies in the same industry in the same period, denoted as *Comp Acct$_{i,t}$*, to measure the comparability of company's accounting information. The larger the value of *Comp Acct$_{i,t}$*, the better the comparability of the company's accounting information.

According to the comparability of accounting information, our process of grouping the research samples is as follows. First, we sort the comparability level of accounting information of enterprises according to industry and year from small to large. Second, we divided the sorted samples into three groups based on one-third quantiles and two-thirds quantiles. Third, we use the first third group of samples and the last third group of samples to conduct group regression tests respectively. We compare the differences in the level of comparability of accounting information and the impact of multilingual disclosure of ESG reports on the enthusiasm of foreign investors.

In columns (1) and (2) of Table 10, there is no significant correlation between the multiplication term *Treat×Post* and the variable *Shareholdmun*. In column (3), the intersection multiplication term *Treat×Post* is significantly related to the variable *ΔSharehold* at the 1% level. In column (4), there is no significant correlation between the intersection multiplication term *Treat×Post* and the variable *ΔSharehold*. In the all-Chinese context, the comparability of

**Table 10. Impact of differences in the comparability of accounting information.**

| VARIABLES | (1) | (2) | (3) | (4) |
|---|---|---|---|---|
| | The comparability of accounting information is low | The comparability of accounting information is high | The comparability of accounting information is low | The comparability of accounting information is high |
| | Shareholdmun | | ΔSharehold | |
| Treat×Post | 0.2231 | 0.5164 | 30.0801*** | 22.5671 |
| | (0.3654) | (0.8656) | (3.0980) | (1.3649) |
| Size | -0.1596 | -0.0274 | 2.6771*** | 3.0683*** |
| | (-1.3368) | (-0.5351) | (3.3948) | (2.6215) |
| Age | 0.7245 | 0.4652*** | -20.0581*** | -20.0723*** |
| | (1.2469) | (2.6226) | (-5.1982) | (-4.8825) |
| Roa | 2.0556 | -0.9962 | -4.8071 | -19.9901 |
| | (1.4654) | (-1.0732) | (-0.5112) | (-0.9337) |
| Lev | 1.4901 | 1.4682*** | -4.6013 | -23.4596** |
| | (1.6329) | (3.3107) | (-0.7673) | (-2.3139) |
| Lon | -1.9131 | 0.2778 | 29.5091*** | 6.6256 |
| | (-1.0983) | (0.3158) | (2.6163) | (0.3278) |
| Top1 | -0.0043 | 0.0004 | -0.2802*** | 0.3556*** |
| | (-0.4033) | (0.0926) | (-3.9700) | (3.9561) |
| Soe | 0.8631* | -0.3022* | 2.4311 | -11.573*** |
| | (1.8893) | (-1.9460) | (0.8187) | (-3.2621) |
| LnGDP | 0.0312 | -0.0258 | -1.0521** | -0.0222 |
| | (0.3420) | (-1.4308) | (-2.1264) | (-0.0539) |
| City_employ | 0.0186 | 0.1078 | -9.9261*** | -5.6162 |
| | (0.0441) | (0.6565) | (-3.3072) | (-1.4026) |
| City_RD | 0.2052 | 0.1973** | 0.9264 | 5.6863** |
| | (0.7078) | (2.0079) | (0.8041) | (2.3746) |
| Area | Yes | Yes | Yes | Yes |
| Techind | Yes | Yes | Yes | Yes |
| Year | Yes | Yes | Yes | Yes |
| Firm | Yes | Yes | Yes | Yes |
| Constant | 1.8921 | -1.0610 | 9.7245*** | 5.2442 |
| | (0.0462) | (-0.6747) | (3.3316) | (1.3776) |
| Observations | 174 | 391 | 174 | 391 |
| R-squared | 0.2391 | 0.1638 | 0.3821 | 0.4089 |
| Coefficient difference test | Prob > chi2 = 0.0573 (Treat×Post) | | Prob > chi2 = 0.0013 (Treat×Post) | |

Note: We report t-statistics based on robust standard errors clustered by firm in parentheses, while ***, **, * denote statistical significance at 1%, 5%, and 10%levels, respectively.

corporate information is low, and corporate annual report information does not have information transparency for foreign investors. Multilingual ESG report information disclosure strengthens the transparency of non-financial information, is complementary to foreign investors' understanding of corporate financial information, enhances the transparency and readability of corporate financial information, and attracts foreign investors to expand their shareholdings.

**6.2.4 The impact of the small and medium investor service center system.** After the reform and opening up policy, China's market economy has developed rapidly, and the legal

system for regulating the quality of information disclosure in the capital market has gradually improved. China's regulatory system reduces the space for corruption and other fraud, but is not associated with better public or private product quality [62], and it creates at least two problems. First, it greatly increases the manager's paperwork process and hinders the manager's efficiency and enthusiasm. Procedural formalism is systematically greater in civil law countries, bringing the costs of longer procedures but not better justice, and higher formalism is associated with lower quality legal system quality. Second, some provisions are too specific and can remove managers' discretion to make decisions based on local circumstances. Managers' cautious attitude is very important to China's development, and extensive regulations and inspection systems make China's political structure more centralized, which may harm economic growth in the long run [63]. In 2014, the China Securities Regulatory Commission decided to establish a service center for small and medium-sized investors. Its purpose is to protect the right to know of small and medium-sized investors, improve the voting mechanism and compensation mechanism, strengthen and improve the protection system for small and medium-sized investors, and resolve the disputes between investors and listed companies and other information disclosure entities civil disputes. According to laws and regulations, the Small and Medium Investor Service Center is given the right to inspect, advise, question, and vote shareholders. The Small and Medium Investor Service Center participates in shareholder meetings, major asset restructuring briefings, investor briefings, performance briefings, etc., and makes public statements, online inquiries and other methods to exercise rights.

The improvement of institutions in the new era is accompanied by an increase in formation procedures [64]. It is one of the important responsibilities of the service center for small and medium-sized investors to accept the entrustment of small and medium-sized investors and provide mediation and settlement services for securities and futures disputes. Small and medium-sized investors should pay attention to the solicitation announcement issued by the investment service center, submit relevant materials within the solicitation period, and sign on the litigation legal documents. The excessive specific provisions for investors to apply for the right to exercise the Small and Medium Investor Service Center may affect investors' decision-making power on effective information disclosure. The disclosure of ESG reports by enterprises in Chinese and English will further enhance the transparency of information disclosure by Chinese enterprises to investors in international capital markets. Compared to obtaining investment decision information through the firm's ESG report, the investor's exercise through the Small and Medium Investor Service Center may increase the cost of the formal process.The Small and Medium Investor Service Center participates in the implementation of the company's rights exercise system, and it is possible for companies to use multiple languages to disclose the effect of ESG reports on foreign investors.

We use dummy variables to measure the exercise of the Small and Medium Investors Service Center. If the Small and Medium Investors Service Center participates in the media briefing, public voice, online inquiry, etc., we define the exercise of the Small and Medium Investors Service Center and assign *Exercise* = 1. Otherwise, we assign *Exercise* = 0. This paper divides the research sample into two groups according to the participation of Small and Medium Investor Service Center in the exercise of corporate rights, and performs group regression to test the difference in the exercise of rights by Small and Medium Investor Service Center, and the impact of multilingual disclosure of ESG reports on the enthusiasm of foreign investors.

In column (1) of Table 11, there is no significant correlation between the cross product *Treat×Post* and the variable *Shareholdmun*. In column (2), the cross product *Treat×Post* is significantly positively correlated with the variable *Shareholdmun* at the 5% level. In column (3), the cross product *Treat×Post* is significantly positively correlated with the variable *ΔSharehold*

**Table 11. Impact results of the exercise of rights by the Small and Medium Investor Service Center.**

| VARIABLES | (1) | (2) | (3) | (4) |
|---|---|---|---|---|
| | Small and medium investors service center exercise rights (Exercise = 1) | Small and medium investors service center not exercise rights (Exercise = 0) | Small and medium investors service center exercise rights (Exercise = 1) | Small and medium investors service center not exercise rights (Exercise = 0) |
| | *Shareholdmun* | | *ΔSharehold* | |
| Treat×Post | 0.1609 | 0.8171** | 13.6671*** | 23.4673*** |
| | (0.9116) | (2.2217) | (3.1686) | (5.1672) |
| Size | 0.2578*** | -0.1589** | 2.8456 | 1.0061 |
| | (3.3142) | (-2.0417) | (1.1425) | (1.0828) |
| Age | -1.5956*** | 0.4041 | -24.8321*** | -11.1411*** |
| | (-5.4799) | (1.2259) | (-3.4358) | (-2.7238) |
| Roa | -2.0736** | 1.3245 | -4.0367 | 0.5609 |
| | (-2.0508) | (0.9427) | (-1.3144) | (0.0501) |
| Lev | -1.0101* | 1.9656*** | 15.6671 | -2.29782 |
| | (-1.7613) | (3.0103) | (0.8584) | (-0.3248) |
| Lon | -1.1577 | -2.3901* | -9.4589 | 21.7293 |
| | (-0.6831) | (-1.8870) | (-1.6089) | (1.6093) |
| Top1 | 0.0047 | -0.0114 | 0.9638*** | 0.3367*** |
| | (0.9086) | (-1.4784) | (5.8117) | (3.0357) |
| Soe | -0.7721*** | 0.5212* | -29.0011*** | -15.6192*** |
| | (-4.4235) | (1.8824) | (-5.2556) | (-5.4611) |
| LnGDP | 0.0069 | -0.1491*** | 0.1578 | 0.2670 |
| | (0.3875) | (-3.9171) | (0.2980) | (0.4397) |
| City_employ | 0.0979 | -1.3002 | -1.4521 | -21.7789** |
| | (0.4705) | (-1.5008) | (-0.1933) | (-2.0537) |
| City_RD | 0.0901 | -1.6045 | 3.2391 | -18.5251* |
| | (0.7290) | (-1.4022) | (1.1866) | (-1.6871) |
| Area | Yes | Yes | Yes | Yes |
| Techind | Yes | Yes | Yes | Yes |
| Year | Yes | Yes | Yes | Yes |
| Firm | Yes | Yes | Yes | Yes |
| Constant | -8.9143 | 12.6789 | 10.9891 | 2.1362** |
| | (-0.4268) | (1.4986) | (0.1434) | (2.0594) |
| Observations | 210 | 355 | 210 | 355 |
| R-squared | 0.6618 | 0.1101 | 0.9145 | 0.2646 |
| Coefficient difference test | Prob > chi2 = 0.0002 (Treat×Post) | | Prob > chi2 = 0.0000 (Treat×Post) | |

Note: We report t-statistics based on robust standard errors clustered by firm in parentheses, while ***, **, * denote statistical significance at 1%, 5%, and 10%levels, respectively.

at the 1% level. In column (4), the cross product *Treat×Post* is significantly positively correlated with the variable *ΔSharehold* at the 1% level. The Small and Medium Investors Service Center does not have the right to exercise, and the Small and Medium Investors Service Center does not have the right to participate in corporate shareholder meetings and major asset restructuring media briefings, public voice, and online inquiry on behalf of small and medium investors. Multilingual disclosure of ESG reports by enterprises is an important way to improve information transparency and ensure the quality of investors' information. Foreign

investors' understanding of corporate ESG information will further enhance their ability to understand corporate financial information. After the information needs of foreign investors are guaranteed, the enthusiasm of holding Chinese enterprises will be enhanced. Theoretically speaking, the dual guarantee of the implementation of the rights exercise by the Small and Medium Investors Service Center and the use of multilingual disclosure of ESG reports by enterprises will further enhance the enthusiasm of foreign shareholding in Chinese enterprises. The test results show that, compared with the situation where the Small and Medium Investors Service Center does not exercise the right, the implementation of the Small and Medium Investors Service Center has a weaker impact on foreign investors by making enterprises disclose ESG reports in multiple languages. The construction of the system to protect the rights and interests of central investors and the increased formalism cost will weaken the impact of multilingual disclosure of ESG reports on the enthusiasm of foreign shareholding.

**6.2.5 The impact of regional cultural integration.**   The development of China's capital market has benefited from the rapid development of transportation. The opening of China's high-speed rail has accelerated the mobility of human resources in various regions and promoted the spread of different regional cultures. After the opening of China's high-speed rail, the integration of cultures in different regions has reduced the barriers to dialect communication between people in different regions and promoted the unification of language in communication among people in various regions. The frequent use of Mandarin and English has become a major trend in language and culture. China has a vast territory. Although China's high-speed rail lines are spread across all provinces, there are obvious differences in the convenience of opening high-speed rail lines in each province, and there are significant differences in the cultural integration of different provincial-level regions. On the basis of high-speed rail connections between provinces, there are significant differences in the high-speed rail connections between prefecture-level cities in various provinces and regions. Interoperable high-speed rail between prefecture-level cities will speed up the speed and degree of cultural integration within the region, promote the familiarity of people in the same province using Mandarin and English, and further promote the openness of local people's language and culture to the outside world. The accessibility of language communication facilitates the spread of ideas while enhancing the transparency of information.

Culture subtly affects the way corporate management discloses information. Language is an important way of cultural communication. Corporate ESG reports are the communication language for corporate sustainable development cultural concepts. Corporate ESG information disclosure represents corporate ESG cultural concepts. In provinces with high-speed rail connections between prefecture-level cities, the convenience of transportation has expanded the mobility of business managers and market investors. The management concept of sustainable corporate development and the concept of ESG investment in the capital market have spread with the convenience of high-speed rail operation, and have become capital market information that people communicate with every day. In situations where regional cultural integration is slow, written reporting language is an important channel for spreading culture. ESG reports are an important carrier for the capital market to communicate the concept and culture of sustainable development. Only by diversifying the language of ESG reports disclosed by companies can they meet the needs of different types of investor cultures and speed up the spread of corporate ESG concepts to the capital market. Differences in the degree of regional cultural integration affect the effectiveness of companies using multiple languages to disclose ESG reports to attract foreign investors.

China has 34 provincial-level administrative regions, nine of which have high-speed rail connections between prefecture-level cities. In December 2018, Fujian Province realized high-speed rail connections between prefecture-level cities. In December 2019, Anhui Province,

Jiangsu Province, Jiangxi Province, and Hebei Province realized high-speed rail connections between prefecture-level cities. In December 2021, Guangdong Province realized high-speed rail connections between prefecture-level cities. In June 2022, Hunan Province and Henan Province will realize interoperable high-speed rail between prefecture-level cities, and in August 2023, Guangxi Province will realize interoperable high-speed rail between prefecture-level cities. The opening of high-speed rail is an important criterion for promoting regional cultural integration. We use the situation of high-speed rail interconnection between prefecture-level cities in the area where the company is located to measure the degree of regional cultural integration. The effect of the opening of the high-speed railway on promoting cultural exchanges has a lag, and the effect will start to be felt the following year after the opening of the high-speed railway at the end of the year. If the region where the enterprise is located is a province that realizes high-speed rail connectivity between prefecture-level cities in December 2018, December 2019 and December 2021, the regional dummy variable is assigned a value of 1 from 2019, 2020 and 2021 respectively, and the region where the enterprise is located does not realize high-speed rail connectivity between prefecture-level cities, the value of the region dummy variable is 0. During the time period of the research sample of this paper, there are eight areas where the companies are located that have high-speed rail connections between prefecture-level cities. The research samples are divided into two groups according to whether the provincial-level region where the enterprise is located is interconnecting with prefecture-level cities by high-speed rail. Grouped regression is performed to test regional cultural integration. Different situations and multilingual disclosure of ESG reports affect the enthusiasm of foreign investors.

In column (1) of Table 12, there is no significant correlation between the cross-multiplication term *Treat × Post* and the variable *Shareholdmun*. The intersection multiplication term *Treat×Post* in column (2) has a significant positive correlation with the variable *Shareholdmun* at the 1% level. The intersection multiplication term *Treat×Post* in column (3) has a significant positive correlation with the variable *ΔSharehold* at the 10% level. The intersection multiplication term *Treat×Post* in column (4) has a significant positive correlation with the variable *ΔSharehold* at the 1% level. The degree of regional cultural integration is low, and the spread of corporate ESG concepts is low. Multilingual ESG report information disclosure expands external communication channels for corporate ESG concepts and enhances the transparency of corporate ESG responsibility information, which helps attract the number of foreign investors and expand the scale of overseas capital holdings.

## 7. Conclusions and discussion

### 7.1 Conclusions

To build a Chinese enterprise into a world-class enterprise, it is necessary to combine the "internal circulation" of the domestic capital market and the "external circulation" of the international capital market. Chinese enterprises attract more overseas capital to participate in the domestic capital market and achieve the combination of "external circulation" and external circulation in the capital market, which is conducive to the sustainable development of Chinese enterprises. Information on the sustainable development of Chinese enterprises needs to be disseminated to the international capital market to promote understanding among foreign investors.

Based on a quasi-natural experiment in which Chinese companies disclose English versions of their ESG reports, we uses a multi-period difference-in-differences model (DID) to study the impact of multilingual ESG report disclosure on the enthusiasm of foreign investors. The study find that Chinese companies' choice to disclose ESG reports in English has a stimulating

**Table 12. The impact of differences in regional cultural integration.**

| VARIABLES | (1) | (2) | (3) | (4) |
|---|---|---|---|---|
| | Regional cultural integration is high | Regional cultural integration is low | Regional cultural integration is high | Regional cultural integration is low |
| | *Shareholdmun* | | *ΔSharehold* | |
| Treat×Post | 0.1109 | 2.6689*** | 1.7931* | 8.8232*** |
| | (0.9921) | (8.6867) | (1.6962) | (10.1361) |
| Size | -0.0328 | -1.4122*** | -0.9089 | -3.8812*** |
| | (-0.2070) | (-9.3205) | (-0.5854) | (-9.0458) |
| Age | 0.5343 | -5.1742*** | -7.2071 | 17.4821*** |
| | (0.9847) | (-3.6312) | (-1.1441) | (4.3301) |
| Roa | 1.4674 | 1.9491*** | 4.8372 | 2.4487 |
| | (0.8579) | (3.1881) | (0.3314) | (1.4138) |
| Lev | 0.6778 | 10.5001*** | -1.8767 | -8.5421*** |
| | (0.6485) | (13.8961) | (-0.1803) | (-3.9912) |
| Lon | -1.2890 | -4.9251*** | 34.3677 | 30.9364*** |
| | (-0.6596) | (-4.4458) | (1.5419) | (9.8618) |
| Top1 | -0.0161 | -0.1761*** | 0.4412** | -0.0053 |
| | (-1.1739) | (-7.7122) | (2.4401) | (-0.0812) |
| Soe | 0.8478* | 0.6942 | -15.4501*** | -2.8889 |
| | (1.7400) | (0.3190) | (-3.0647) | (-0.2218) |
| LnGDP | -0.2541 | 0.0088 | 2.0455 | -0.2221*** |
| | (-0.3471) | (0.5403) | (0.4149) | (-4.8157) |
| City_employ | 2.5662 | -1.0451*** | -22.4798 | 5.7822*** |
| | (0.7777) | (-2.9944) | (-1.0746) | (5.8496) |
| City_RD | 1.7373 | 1.7241* | 4.6178 | -5.3339** |
| | (1.5659) | (1.9208) | (1.4521) | (-2.1000) |
| Area | Yes | Yes | Yes | Yes |
| Techind | Yes | Yes | Yes | Yes |
| Year | Yes | Yes | Yes | Yes |
| Firm | Yes | Yes | Yes | Yes |
| Constant | -2.4921 | 1.4922*** | 2.1791 | -4.6215*** |
| | (-0.7981) | (4.6174) | (1.0902) | (-5.0542) |
| Observations | 133 | 432 | 133 | 432 |
| R-squared | 0.1008 | 0.7637 | 0.2656 | 0.7256 |
| Coefficient difference test | Prob > chi2 = 0.0103 (Treat×Post) | | Prob > chi2 = 0.0000 (Treat×Post) | |

Note: We report t-statistics based on robust standard errors clustered by firm in parentheses, while ***, **, * denote statistical significance at 1%, 5%, and 10%levels, respectively.

effect on the enthusiasm of foreign investors for shareholding, which is mainly reflected in the expansion of the scale of foreign shareholding and the increase in the number of shareholders. Further research show that multilingual ESG report disclosure enhances foreign investors' understanding of company annual reports. The employment of the big four international accounting firms by companies to audit financial reports, analyst attention and comparability of accounting information affect the incentive effect of multilingual ESG report disclosure on foreign investors. The protection system for small and medium investors and regional cultural integration affect the effect of Chinese enterprises' disclosure of English ESG reports on foreign investors.

## 7.2 Further discussion

This paper makes important implications to both theory and practice. Theoretically, our study contributes to previous literature in the following ways.

First, this research contributes to the existing literature on ESG information disclosure by exploring how companies' disclosure of ESG reports in multiple languages will affect the enthusiasm of foreign investors for holding shares. The research proposes that companies using multiple languages to disclose ESG reports can effectively increase the enthusiasm of foreign capital holdings, which is specifically reflected in the increase in the scale of foreign capital holdings and the increase in the number of foreign shareholders. Our research provides a new perspective on ESG information disclosure by highlighting the role of ESG reporting in multiple languages. The impact of corporate ESG rating levels on financial management and corporate governance has been discussed in a large amount of literature. This article focuses on the impact of ESG reporting information disclosure methods on foreign corporate investors. Our research shows that analyst attention and accounting information comparability are poor, and companies that hire non-Big Four auditing firms to audit financial reports are more enthusiastic about foreign shareholding disclosures in multilingual ESG reports. A large amount of research literature shows that ESG ratings affect shareholder governance and investor decision-making [65]. Our findings extend important perspectives in previous literature. We research the impact of companies on the interests of external investors from the perspective of ESG information disclosure, and improve the decision-making usefulness of Chinese companies' ESG report disclosures in the international market.

Second, this research provides a new quasi-experimental scenario to explore the mechanism of the influence of ESG report disclosure language on cross-border capital flows, and helps to solve the endogenous causality of corporate governance behavior on ESG information disclosure in the existing literature. ESG information disclosure has become a broad consensus in the international capital market. The existing literature on corporate ESG metrics includes ESG ratings by ESG professional institutions, ESG performance scores and whether ESG report information is disclosed to explore the impact of corporate ESG performance on corporate decisions and behaviors [66,67]. Some scholars also try to reflect their alliances and cooperative actions from the common preferences of institutional investors for ESG, and clusters of ESG interests among institutional investors will work together to convince portfolio companies to innovate in low carbon [68]. However, the ESG performance of Chinese listed companies and the common ESG preferences of institutional investors are not an effective way to promote Chinese companies to convey ESG information to overseas investors. This paper identifies the information transmission methods that drive foreign investment into Chinese companies based on a quasi-natural experiment using a company to disclose ESG reporting information in multiple languages. Our research provides a useful roadmap for Chinese enterprises to transfer the performance of ESG responsibilities to the international capital market, which deserves more exploration in the following study.

Third, this research expands the existing literature on the functional effects of ESG information by exploring how companies use English to disclose ESG reports to improve the readability of corporate annual report information for foreign investors. Enterprises' fulfillment of ESG responsibilities is an important way to promote high-quality economic development. However, the existing literature uses the evaluation indicators of ESG professional rating agencies to study the interaction between Chinese overseas investors and is very limited. This research find that the use of enterprises to disclose ESG reports in English has the enthusiasm of foreign investors to understand the information of corporate annual reports, which is more significant in the poor attention of analysts and the comparability of accounting information,

and the enterprises that employ non-big four accounting firms to audit financial reports. This result is consistent with the popular view in the literature that annual report information readability affects capital market investor decisions [69,70]. This means that the information disclosure of the English version of ESG reports of Chinese listed companies is an effective means to reduce information asymmetry among foreign investors.

In fact, these findings have some implications for Chinese companies. Based on the above research conclusions, we draw the following three revelations. First, Chinese enterprises should strengthen the international versatility of information disclosure. Chinese companies should expand their attention from domestic investors to international investors based on the opening up of the Chinese market. Sustainable development goals are a topic of common concern to the international community. Chinese companies that use Chinese language to disclose operating reports only use the domestic capital market. To attract foreign investment, Chinese companies need to expand the international versatility of the company's report disclosure language.

Second, Chinese companies should strengthen international exchanges of ESG concepts. The sustainable development of Chinese enterprises is a topic of common concern in the international capital market. How foreign investors understand the ESG concepts and implementation of ESG responsibilities of Chinese enterprises is an important issue. This is also a key issue in promoting international capital circulation in China's capital market. Chinese companies' use of multiple languages to disclose ESG in Chinese and English will help strengthen the integration of corporate ESG information into the international capital market, promote Chinese foreign investors' understanding and recognition of Chinese companies' ESG information, and encourage foreign investors to be enthusiastic about holding shares in Chinese companies.

Third, Chinese companies should consider the difference in effectiveness of the English version of the ESG report. If companies disclose their ESG reports in English, it will help enhance the readability of the company's annual reports for foreign investors and alleviate the information asymmetry between foreign investors and companies. The positive effect of the English version of ESG reports on promoting foreign investors' shareholdings may not always be satisfactory. As mentioned in this paper, the comparability of corporate accounting information and the supervision effect of external analysts and auditors can enhance the effect of the English version of the ESG report. The English version of ESG reports disclosed by Chinese companies makes up for the readability of annual report information and is an important mechanism to increase foreign shareholdings. Therefore, improving the readability of information disclosure by Chinese enterprises is the key to promoting the flow of capital within and outside China.

## 8.Limitation and future research

This paper studies the impact of multilingual disclosure of ESG reports on the enthusiasm of foreign ownership. There are two limitations: firstly, this paper has limitations in the measurement of research variables. Within a fixed period, the degree of transaction volume completed in a short period of time can reflect the enthusiasm of foreign investment. Since it is impossible to observe the daily transaction time series of Chinese companies held by foreign investors, this paper uses the scale of domestic and foreign shareholdings in an accounting period and the number of foreign shareholders as proxy variables to measure the enthusiasm of foreign investors. Secondly, this paper has limitations in the research method. There are no unified standards for the disclosure content of corporate ESG reports, and there are no unified standards for the content of corporate ESG reports disclosed in multiple languages. This paper sets

up dummy variables based on whether companies disclose ESG reports in Chinese and English, and uses a quasi-natural experiment to conduct empirical research as a preliminary exploration. After ESG reports are unified and standardized, the measurement method of ESG reports disclosed in multiple languages needs to be further improved.

There are more opportunities for future research on the topic of the impact of multilingual disclosure of ESG reports on the enthusiasm of foreign investors for holding shares. This paper puts forward research prospects in the following three aspects: firstly, establish a unified standard for the disclosure of corporate ESG reports. We can use text analysis to study the impact of the comparability of ESG information disclosed by companies in multiple languages on the behavioral decisions of foreign investors. Secondly, China's capital market is increasingly open to the world. After the information disclosure mechanism for foreign investors' capital investment in China has been improved and matured, we further used survey research methods to study the impact of Chinese companies' ESG information disclosure on the enthusiasm of foreign investors for holding shares. Thirdly, with the gradual improvement of corporate ESG report disclosure methods, the Chinese capital market has improved its mechanism for disclosing listed company information in multiple languages to the international market. The study of multilingual disclosure of ESG reports by enterprises in international capital market debt financing, import and export trade, and other aspects of research needs to be explored.

## Supporting information

**S1 Appendix.**
(DOCX)

## Author Contributions

**Data curation:** Ruixue Bao.

**Formal analysis:** Ruixue Bao.

**Methodology:** Ruixue Bao.

**Writing – original draft:** Ruixue Bao, Li Wei.

**Writing – review & editing:** Li Wei.

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
