## [Decision Letter · Decision Letter 0]

17 Nov 2023

PONE-D-23-30913The impact of companies disclosing ESG reports in multiple languages on the enthusiasm of foreign investors for holding sharesPLOS ONE

Dear Dr. Bao,

Thank you for submitting your manuscript to PLOS ONE. After careful consideration, we feel that it has merit but does not fully meet PLOS ONE’s publication criteria as it currently stands. Therefore, we invite you to submit a revised version of the manuscript that addresses the points raised during the review process.

**ACADEMIC EDITOR: **

During the review process, recommendations for major revision of the paper were expressed. I agree with the reviewers that this paper needs significant improvement and provided myself additional feedback after evaluation of the manuscript. The reviewers have recommended a number of citations as a part of their review. I recommend that you thoroughly evaluate these requested references and determine whether the articles are relevant to the current study. You may feel free to disregard references with tangible relevance to the study reported in the manuscript. 

I invite the authors to revise the paper after careful consideration of my comments and the recommendations of the reviewers. For publication consideration, pleases provide full responses to all the bellow comments.

We look forward to receiving your revised manuscript.

Kind regards,

Ionela Munteanu, PhD

Academic Editor

PLOS ONE

4. PLOS requires an ORCID iD for the corresponding author in Editorial Manager on papers submitted after December 6th, 2016. Please ensure that you have an ORCID iD and that it is validated in Editorial Manager. To do this, go to ‘Update my Information’ (in the upper left-hand corner of the main menu), and click on the Fetch/Validate link next to the ORCID field. This will take you to the ORCID site and allow you to create a new iD or authenticate a pre-existing iD in Editorial Manager. Please see the following video for instructions on linking an ORCID iD to your Editorial Manager account: https://www.youtube.com/watch?v=_xcclfuvtxQ.

Reviewers' comments:

Reviewer's Responses to Questions

**Comments to the Author**

1. Is the manuscript technically sound, and do the data support the conclusions?

Reviewer #1: Partly

Reviewer #2: Yes

Reviewer #3: Partly

2. Has the statistical analysis been performed appropriately and rigorously? 

Reviewer #1: I Don't Know

Reviewer #2: Yes

Reviewer #3: Yes

3. Have the authors made all data underlying the findings in their manuscript fully available?

Reviewer #1: No

Reviewer #2: No

Reviewer #3: No

4. Is the manuscript presented in an intelligible fashion and written in standard English?

Reviewer #1: No

Reviewer #2: Yes

Reviewer #3: No

5. Review Comments to the Author

Reviewer #1: The authors employ the multi-period difference-in-difference model (DID) to investigate the impact of multilingual ESG report disclosure on the enthusiasm of foreign investors. They discover that when Chinese companies disclose ESG reports in both Chinese and English, it enhances the enthusiasm of foreign investors to hold shares. This enhancement is chiefly evident in the expansion of the company's foreign shareholding quota and the increase in the number of shareholders. Subsequent research reveals that the disclosure of multilingual ESG reports addresses the readability concerns of company annual reports for foreign investors. I have the following comments for the authors to further improve the paper:

1. The paper needs to conform to the format standard of PLOS ONE. Now it reads like a submission to an economics filed journal.

2. Currently, the literature review is not complete. There are several recent, worth-dicussing papers that are currently missing from the paper. These include, but not limited to the following: Corporate sustainability policies and corporate investment efficiency: Evidence from the quasi-natural experiment in China (by Kung-Cheng Ho, Cheng Yan, Zhicheng Mao, and Jiafu An, in Energy Economics), in which the authors look at a similar topic, ESG, in the Chinese setting.

3. When doing the analysis, the authors may also consider the quality of local institutions in moderating the baseline results. For example, does local formalism has any bearings on the discovered relationship. "China’s rule of law in New Era: The rise of regulation and formalism" (Jiafu An, Wenxuan Hou and Yun Zhang, Journal of Chinese Economic and Business Studies. 2019) provides some institutional backgound and should be discussed when conducting the analysis.

4. How about local culture and its role in shaping the baseline relationship? Since language is a reflection of culture, I think culture should also play a role in the baseline finding. Before the authors actuall do the heterogenous analysis based on culture, they need to motivate the analysis base on existing studies on culture and firm outcomes. A few examples can be discussed and followed:

An, J. (2020). Is there an employee-based gender gap in informal financial markets? International evidence. Journal of Corporate Finance, 65, 101737.

An, J., Hou, W., & Lin, C. (2022). Epidemic disease and financial development. Journal of Financial Economics, 143(1), 332-358.

Xu, J., Chen, J., Jiang, M., & An, J. (2023). Inherited trust and informal finance. Journal of Business Finance & Accounting.

An, J., Jiang, M., & Xu, J. (2021). Professional norms and risk-taking of bank employees: Do expectations of peers’ risk preferences matter?. Journal of Financial Stability, 56, 100938.

5. Be sure to the follow the format requirement regarding the paper and tables of PLOS One.

Reviewer #2: The manuscript addresses a very interesting topic, which is why I recommend its publication after making some adjustments. The ideas are well presented, scientifically argued and presented in a logical order. The main adjustments I recommend are:

1. The authors must specify in the abstract the analysed period.

2. The authors should better justify the choice of the analysed period.

3. The authors should better justify the choice of method.

4. The authors must insert a discussions section. In the discussions section, the authors must be preset the results of the study in the context of similar papers. In this section, additional references can used (https://doi.org/10.1007/s43039-022-00056-x;
https://doi.org/10.3390/su13168876)

5. The authors must insert in the final section, considerations regarding limitations of the research and future directions for the research.

Reviewer #3: 1. I suggest to elaborate on the motivation of group variables choice and revise the hypotheses of the research to match the methodological process. Link the hypotheses with the discussions section of the paper and with the conclusions.

2. Does the DID estimator appear in any equation? If yes, a proper identification and description should be provided. In the current form of the paper, the theoretical discussions of the results do not seem to be clearly linked to the description of equations.

3. The explanations to all equations should be revised or completed so that explanations are provided for all used symbols. For example, equations (3) and (4) use regression coefficients β1 to β17, but the explanations in lines 453-456 refer to coefficients β1 to β3 and, additionally, refer to dummy variables S years and M years which are not present in the corresponding equations. Equation (11) needs additional explanations like why subtraction operation is used for different companies accounting systems and what t=15 and 1/16 stand for. In line 642, coefficient is mentioned with reference to model (8), but model (8) does not include such coefficient. Please check and correct.

4. Explain the meaning of numbers in parentheses in Tables 4, 5, 6, 7, 8, 9, 10 preferably at the bottom of each table.

5. Please check the meaning of the following phrases and revise them:

- lines 32-34 (”Financial performance reporting, which will help foreign investors better understand the company’s operations.”),

- lines 539-540 (”They provide investors with incremental information by asking questions and observing management’s tone and tone.”),

- lines 596-597 (”This paper uses whether the big four international accounting firms are hired for corporate financial report auditing to measure the audit status of the big four international auditing firms.”)

6. Explain the meaning of used abbreviation the first time you use them: PCAOB in line 194, CSMAR, CNRDS in line 310, ST and ST* in line 312, PSM in line 315. Also, explain the meaning of the word ”tertiles” used in several parts of the paper.

7. Avoid using capital letters for ”This paper” within the phrase and correct lines 306 and 308.

8. According to subsection 4.1., data was collected from CSMAR database and CNRDS database, but the Data availability statement mentions that the used data are owned by CSMAR and CICPA. Please explain.

6. PLOS authors have the option to publish the peer review history of their article (what does this mean?). If published, this will include your full peer review and any attached files.

Reviewer #1: No

Reviewer #2: No

Reviewer #3: **Yes: **Ionela Munteanu, PhD

---

## [Author Response · Author response to Decision Letter 0]

27 Dec 2023

Dear editor,

Our most sincere thanks to you, Reviewer 1, Reviewer 2, and Reviewer 3 for the very thoughtful review and helpful comments, and suggestions on our writing in this round.

Based on the suggestions of reviewer 1 and reviewer 3, we have revised the literature review and hypothesis discussion sections of the paper to highlight the theoretical logic of the paper. We also supplemented the empirical test of the paper, and modified the writing details and format to improve the quality of the paper. With your thoughtful comments, our latest version has improved a lot.

Below, we provide the original comments from Reviewer 1, Reviewer 2 and Reviewer 3 in boldface and our responses to each item in regular font. We do this point by point. We highlight the main amendment in blue in the revised manuscript.

Response to Reviewer 1

（PONE-D-23-30913）

1.The paper needs to conform to the format standard of PLOS ONE. Now it reads like a submission to an economics filed journal.

 Thanks for your helpful suggestion.

According to the format specification provided by Plos One editor, we modified the format of the full text. The format specifications for Plos One are as follows:

2.Currently, the literature review is not complete. There are several recent, worth-dicussing papers that are currently missing from the paper. These include, but not limited to the following: Corporate sustainability policies and corporate investment efficiency: Evidence from the quasi-natural experiment in China (by Kung-Cheng Ho, Cheng Yan, Zhicheng Mao, and Jiafu An, in Energy Economics), in which the authors look at a similar topic, ESG, in the Chinese setting.

 Thanks for your suggestion.

Inspired by this review, we supplemented the research literature on corporate sustainability with a Chinese setting in the literature review to complete the shortcomings of the literature review. Our added literature review includes research on corporate sustainability policy and corporate investment, climate change risk, ESG rating divergence, and ESG performance. Our additional literature is as follows:

1.Kung-Cheng Ho, Cheng Yan , Zhicheng Mao , Jiafu An.Corporate sustainability policies and corporate investment efficiency: Evidence from the quasi-natural experiment in China.Energy Economics.Forthcoming 2023.

2.Li, Q., H. Shan, Y. Tang, V.Yao.Corporate Climate Risk:Measurements and Responses.SSRN Working Paper.2020; No.3508497.

3.Deqing Luo, Jingzhou Yan, Qianhui Yan. The Duality of ESG: Impact of Ratings and Disagreement on Stock Crash Risk in China. Finance Research Letter. Forthcoming 2023.

4.Emawtee Bissoondoyal-Bheenick, Robert Brooks, Hung Xuan Do. ESG and firm performance: The role of size and media channels. Economic Modelling. Forthcoming 2023.

5.Feng He, Yaqian Feng, Jing Hao. Corporate ESG rating and stock market liquidity: Evidence from China. Economic Modelling. Forthcoming 2023.

We have revised the first paragraph of the literature review “2.1 Research on the impact of ESG information disclosure on corporate performance”, which is as follows:

The idiosyncratic information of enterprises in terms of strategic operations, social interests, and sustainable development is transmitted to investors, which affects the pricing efficiency of the capital market. ESG performance has an information effect. The green disclosure requirements significantly increases corporate investment efficiency[14]. Better-performing companies disclose their sustainability performance in a way that is optimistic, certain, clear and more readable[15]. The higher ESG ratings contribute to a diminished likelihood of stock price crash risk, while the salutary impact is diminished by the prevalent ESG disagreement among rating agencies [16]. Larger companies tend to invest in ESG activities to better reflect the needs of their stakeholders. Companies with better media coverage can reduce information asymmetry among stakeholders in ESG investments[17]. ESG performance and ESG rating events with high financial investment behavior can promote corporate innovation [15]. ESG rating significantly improves stock liquidity [18]. Firms with sustainability requirements tend to have better market performance [19]. Companies reduce agency costs and improve corporate performance by fulfilling social responsibilities [15]. Companies with good ESG performance often have a strong sense of social responsibility, environmental awareness, and a high-level governance mechanism, which helps companies establish a high-quality image in the capital market [20]. Good ESG ratings and performance not only reduce information risks and financing costs, but also improve investment efficiency [21]. Low ESG ratings and negative news increase corporate operating risks, damage corporate value, and even endanger customer reputations, causing customer stock prices to fall [22]. Companies with good ESG performance integrate social responsibility into product differentiation strategies, build brand effects by actively fulfilling social responsibilities to improve reputation, and produce differentiated high-quality products that are different from competitors [23].

3.When doing the analysis, the authors may also consider the quality of local institutions in moderating the baseline results. For example, does local formalism has any bearings on the discovered relationship. "China’s rule of law in New Era: The rise of regulation and formalism" (Jiafu An, Wenxuan Hou and Yun Zhang, Journal of Chinese Economic and Business Studies.2019) provides some institutional backgound and should be discussed when conducting the analysis.

Thanks for your suggestion.

Inspired by this comment, we conducted an Additional examination of the moderating effect of local institutions on the fundamental regression results in the section “6.2 Additional inspection” of our paper. We read the references provided by the reviewers and thought about the process of strengthening supervision in China's scenario system, which may increase the cost of procedural formalism.

We research the impact of multilingual disclosure of ESG reports on foreign investors’ enthusiasm. As for safeguarding the system of small and medium-sized investors, the establishment of the Small and Medium Investor Service Center in China is an important system. The Small and Medium Investor Service Center to participate in corporate shareholder meetings and major asset restructuring media briefing, public voice, online inquiry and other ways to exercise rights. It is one of the important responsibilities of the Small and Medium Investor Service Center to accept the entrustment of small and medium-sized investors and provide mediation and settlement services for securities and futures disputes. Small and medium-sized investors should pay attention to the solicitation announcement issued by the investment service center, submit relevant materials within the solicitation period, and sign on the litigation legal documents. At the same time, investors apply for the rights protection process by entrusting the Small and Medium Investor Service Center, which may also increase the formalism cost of investor procedures. The excessive specific provisions for investors to apply for the exercise of the Small and Medium Investor Service Center may affect investors' decision-making power on effective information disclosure. Compared to obtaining investment decision information through the firm's ESG report, the investor's exercise through the Small and Medium Investor Service Center may increase the cost of the formal process. The participation of the Small and Medium Investor Service Center in the implementation of the enterprise exercise system may make the enterprise use multiple languages to disclose the effect of ESG reports on foreign investors.

In addition, we tested the moderating effect of the central investor service center's participation in corporate governance system on the basic regression results. We use dummy variables to measure the exercise of the Small and Medium Investors Service Center. If the Small and Medium Investors Service Center participates in the media briefing, public voice, online inquiry, etc., we define the exercise of the Small and Medium Investors Service Center and assign Exercise=1. Otherwise, we assign Exercise=0. In this paper, the sample is divided into two groups according to the participation of the Small and Medium Investor Service Center in the exercise of enterprise rights, and the group regression test is carried out to test the differences in the exercise of small and medium investor service centers, and the influence of ESG reports on the enthusiasm of foreign investors is disclosed in multiple languages.

Theoretically speaking, the dual guarantee of the implementation of the rights exercise by the Small and Medium Investors Service Center and the use of multilingual disclosure of ESG reports by enterprises will further enhance the enthusiasm of foreign shareholding in Chinese enterprises. In column (1) of Table 11, there is no significant correlation between the cross product Treat×Post and the variable Shareholdmun. In column (2), the cross product Treat×Post is significantly positively correlated with the variable Shareholdmun at the 5% level. In column (3), the cross product Treat×Post is significantly positively correlated with the variable ΔSharehold at the 1% level. In column (4), the cross product Treat×Post is significantly positively correlated with the variable ΔSharehold at the 1% level. The test results show that, compared with the situation where the Small and Medium Investors Service Center does not exercise the right, the implementation of the Small and Medium Investors Service Center has a weaker impact on foreign investors by making enterprises disclose ESG reports in multiple languages. The construction of the system to protect the rights and interests of central investors and the increased formalism cost will weaken the impact of multilingual disclosure of ESG reports on the enthusiasm of foreign shareholding.

Table 11.Impact results of the exercise of rights by the Small and Medium Investor Service Center.

 （1） （2） （3） （4）

VARIABLES Small and medium investors service center exercise rights Small and medium investors service center not exercise rights Small and medium investors service center exercise rights Small and medium investors service center not exercise rights

 Shareholdmun ΔSharehold

Treat×Post 0.1609 0.8171** 13.6671*** 23.4673***

 (0.9116) (2.2217) (3.1686) (5.1672)

Size 0.2578*** -0.1589** 2.8456 1.0061

 (3.3142) (-2.0417) (1.1425) (1.0828)

Age -1.5956*** 0.4041 -24.8321*** -11.1411***

 (-5.4799) (1.2259) (-3.4358) (-2.7238)

Roa -2.0736** 1.3245 -4.0367 0.5609

 (-2.0508) (0.9427) (-1.3144) (0.0501)

Lev -1.0101* 1.9656*** 15.6671 -2.29782

 (-1.7613) (3.0103) (0.8584) (-0.3248)

Lon -1.1577 -2.3901* -9.4589 21.7293

 (-0.6831) (-1.8870) (-1.6089) (1.6093)

Top1 0.0047 -0.0114 0.9638*** 0.3367***

 (0.9086) (-1.4784) (5.8117) (3.0357)

Soe -0.7721*** 0.5212* -29.0011*** -15.6192***

 (-4.4235) (1.8824) (-5.2556) (-5.4611)

LnGDP 0.0069 -0.1491*** 0.1578 0.2670

 (0.3875) (-3.9171) (0.2980) (0.4397)

City_employ 0.0979 -1.3002 -1.4521 -21.7789**

 (0.4705) (-1.5008) (-0.1933) (-2.0537)

City_RD 0.0901 -1.6045 3.2391 -18.5251*

 (0.7290) (-1.4022) (1.1866) (-1.6571)

Area Yes Yes Yes Yes

Techind Yes Yes Yes Yes

Year Yes Yes Yes Yes

Firm Yes Yes Yes Yes

Constant -8.9143 12.6789 10.9891 2.1362**

 (-0.4268) (1.4986) (0.1434) (2.0594)

Observations 210 355 210 355

R-squared 0.6618 0.1101 0.9145 0.2646

Coefficient difference test Prob > chi2 =0.0002 (Treat×Post) Prob > chi2 =0.0000 (Treat×Post)

Note: We report t-statistics based on robust standard errors clustered by firm in parentheses, while ***, **, * denote statistical significance at 1%, 5%, and 10%levels, respectively.

4. How about local culture and its role in shaping the baseline relationship? Since language is a reflection of culture, I think culture should also play a role in the baseline finding. Before the authors actuall do the heterogenous analysis based on culture, they need to motivate the analysis base on existing studies on culture and firm outcomes. A few examples can be discussed and followed:

An, J. (2020). Is there an employee-based gender gap in informal financial markets? International evidence. Journal of Corporate Finance, 65, 101737.

 An, J., Hou, W., & Lin, C. (2022). Epidemic disease and financial development. Journal of Financial Economics, 143(1), 332-358.

 Xu, J., Chen, J., Jiang, M., & An, J. (2023). Inherited trust and informal finance. Journal of Business Finance & Accounting.

 An, J., Jiang, M., & Xu, J. (2021). Professional norms and risk-taking of bank employees: Do expectations of peers’ risk preferences matter?. Journal of Financial Stability, 56, 100938.

Thanks for your suggestion.

Inspired by this comment, we examined the moderating effect of local culture on the basic regression results in the section “6.2 Additional inspection” of our paper. After reading the references provided by the reviewers, we believe that language is a tool of cultural communication, and regional culture has an important impact on the familiarity of language use. Cultural exchange and integration affect language exchange and capital flow. China’s high-speed rail has promoted the opening and integration of cultures in different regions of China, and provided us with a scene for studying regional cultures.

China is geographically vast. Although China’s high-speed railways are spread across all provinces, there are obvious differences in the convenience of opening high-speed railways among different provinces, and there are significant differences in cultural integration among different provincial regions. On the basis of inter-provincial high-speed rail connectivity, there are significant differences in the inter-provincial high-speed rail connectivity among prefecture-level cities. The high-speed railway between prefecture-level cities accelerates the speed and degree of cultural integration in the region, promotes the familiarity of people in the same province with Mandarin and English languages, and further promotes the openness of local people's language and culture to the outside world. The accessibility of language facilitates the spread of ideas and enhances the transparency of information.

We choose the region where the enterprise is located to achieve high-speed rail connectivity between prefecture-level cities to measure the degree of regional cultural integration. China has 34 provincial-level administrative regions, nine of which have high-speed rail connections between prefecture-level cities. In December 2018, prefecture-level cities in Fujian Province implemented high-speed rail connectivity; in December 2019, prefecture-level cities in Anhui, Jiangsu, Jiangxi and Hebei provinces implemented high-speed rail connectivity; in December 2021, prefecture-level cities in Guangdong Province implemented high-speed rail connectivity; and in June 2022, prefecture-level cities in Hunan and Henan provinces implemented high-speed rail connectivity. In August 2023, Guangxi Province realized high-speed rail connectivity between prefecture-level cities. The effect of the opening of the high-speed railway on promoting cultural exchanges has a lag, and the effect will start to be felt the following year after the opening of the high-speed railway at the end of the year. If the region where the enterprise is located is a province that realizes high-speed rail connectivity between prefecture-level cities in December 2018, December 2019 and December 2021, the regional dummy variable is assigned a value of 1 from 2019, 2020 and 2021 respectively, and the region where the enterprise is located does not realize high-speed rail connectivity between prefecture-level cities, the value of the region dummy variable is 0. In the time interval (2013-2022) of the study samples, 8 enterprises are located in regions that have realized high-speed rail connectivity between prefecture-level cities. The research samples are divided into two groups according to whether the provincial-level region where the enterprise is located is interconnecting with prefecture-level cities by high-speed rail. The group regression test is conducted to test the difference of regional cultural integration, and the influence of ESG report on the enthusiasm of foreign investors is disclosed in multiple languages.

In column (1) of Table 12, there is no significant correlation between the cross-multiplication term Treat×Post and the variable Shareholdmun. The intersection multiplication term Treat×Post in column (2) has a significant positive correlation with the variable Shareholdmun at the 1% level. The intersection multiplication term Treat×Post in column (3) has a significant positive correlation with the variable ΔSharehold at the 10% level. The intersection multiplication term Treat×Post in column (4) has a significant positive correlation with the variable ΔSharehold at the 1% level. The degree of regional cultural integration is low, and the spread of corporate ESG concepts is low. Multilingual ESG report information disclosure expands external communication channels for corporate ESG concepts and enhances the transparency of corporate ESG responsibility information, which helps attract the number of foreign investors and expand the scale of overseas capital holdings.

Table 12. The impact of differences in regional cultural integration.

 （1） （2） （3） （4）

VARIABLES Regional cultural integration is high Regional cultural integration is low Regional cultural integration is high Regional cultural integration is low

 Shareholdmun ΔSharehold

Treat×Post 0.1109 2.6689*** 1.7931* 8.8232***

 (0.9921) (8.6867) (1.6662) (10.1361)

Size -0.0328 -1.4122*** -0.9089 -3.8812***

 (-0.2070) (-9.3205) (-0.5854) (-9.0458)

Age 0.5343 -5.1742*** -7.2071 17.4821***

 (0.9847) (-3.6312) (-1.1441) (4.3301)

Roa 1.4674 1.9491*** 4.8372 2.4487

 (0.8579) (3.1881) (0.3314) (1.4138)

Lev 0.6778 10.5001*** -1.8767 -8.5421***

 (0.6485) (13.8961) (-0.1803) (-3.9912)

Lon -1.2890 -4.9251*** 34.3677 30.9364***

 (-0.6596) (-4.4458) (1.5419) (9.8618)

Top1 -0.0161 -0.1761*** 0.4412** -0.0053

 (-1.1739) (-7.7122) (2.4401) (-0.0812)

Soe 0.8478* 0.6942 -15.4501*** -2.8889

 (1.7400) (0.3190) (-3.0647) (-0.2218)

LnGDP -0.2541 0.0088 2.0455 -0.2221***

 (-0.3471) (0.5403) (0.4149) (-4.8157)

City_employ 2.5662 -1.0451*** -22.4798 5.7822***

 (0.7777) (-2.9944) (-1.0746) (5.8496)

City_RD 1.7373 1.7241* 4.6178 -5.3339**

 (1.5659) (1.9208) (1.4521) (-2.1000)

Area Yes Yes Yes Yes

Techind Yes Yes Yes Yes

Year Yes Yes Yes Yes

Firm Yes Yes Yes Yes

Constant -2.4921 1.4922*** 2.1791 -4.6215***

 (-0.7981) (4.6174) (1.0902) (-5.0542)

Observations 133 432 133 432

R-squared 0.1008 0.7637 0.2656 0.7256

Coefficient difference test Prob > chi2 =0.0103 (Treat×Post) Prob > chi2 =0.0000 (Treat×Post)

Note: We report t-statistics based on robust standard errors clustered by firm in parentheses, while ***, **, * denote statistical significance at 1%, 5%, and 10%levels, respectively.

5.Be sure to the follow the format requirement regarding the paper and tables of PLOS One.

Thanks for your helpful suggestion.

We have revised the full text in strict accordance with the formatting requirements of PLOS One papers and tables.

Response to Reviewer 2

（PONE-D-23-30913）

1. The authors must specify in the abstract the analysed period.

Thanks for your helpful suggestion.

We followed the reviewer's suggestion and supplemented the period of study sample analysis in the first sentence of the abstract. The content of our revised abstract is marked in blue as follows.

Using Chinese A-share listed companies from 2013 to 2022 as samples, we use the multi-period difference-in-difference model (DID) to study the impact of multilingual ESG report disclosure on the enthusiasm of foreign investors. We find that Chinese companies disclose ESG reports in both Chinese and English stimulate the enthusiasm of foreign investors to hold shares. The main manifestations are the expansion of the company's foreign shareholding quota and the increase in the number of shareholders. Further research show that disclosure of multilingual ESG reports makes up for the readability of company annual reports for foreign investors. In the case of companies with poor analyst attention and comparability of accounting information, and companies that hire non-big four auditing firms to audit financial reports, multilingual ESG report disclosures are more positive for foreign shareholdings. The participation of the central investor service center in corporate governance is weak, the degree of regional cultural integration is low, and the disclosure of English ESG reports by Chinese enterprises is conducive to promoting the enthusiasm of foreign shareholding. The research conclusions provide theoretical guidance and empirical reference for enterprises to expand information disclosure methods to foreign investors and attract overseas capital investment.

2. The authors should better justify the choice of the analysed period.

Thanks for your helpful suggestion.

We explained the reasons for selection during sample analysis in detail in the section “4.1 Source and selection of research samples” of the paper. In order to control the influence of deviation on the robustness of test results before and after policy implementation, we select samples of 5 years before and after the policy time point for testing.

In 2018, the China Securities Regulatory Commission revised the Corporate Governance Code for Listed Companies to establish for the first time a corporate ESG information disclosure framework, forcing enterprises to start disclosing social responsibility reports or ESG reports. Since 2018, enterprises have voluntarily chosen to disclose social responsibility reports or ESG reports in multiple languages. In order to maintain the balance between the experimental group and the treatment group, we selected samples 5 years before the experimental event and 5 years after the experimental event. Finally, the sample we selected is China's A-share listed companies from 2013 to 2022.

3. The authors should better justify the choice of method.

Thanks for your helpful suggestion.

We add an explanation of the rationality of the research method selection in section “4.3 Model construction”. In 2018, the China Securities Regulatory Commission revised the “Guidelines for Governance of Listed Companies” to establish a corporate ESG information disclosure framework for the first time, mandating companies to start disclosing social responsibility reports or ESG reports. The establishment of a system for Chinese listed companies to disclose ESG reports has formed a policy basis for companies to disclose ESG reports in multiple languages. This policy basis is an exogenous impact event for companies. In the years after 2018, companies gradually disclosed ESG reports in both Chinese and English. Evaluating policy effects under traditional methods mainly conducts regression tests by setting dummy variables of whether companies disclose ESG reports in English. In comparison, our research sample is panel data, and our research method using the difference-in-differences model model is more scientific, which avoids the problem of reverse causation and more accurately estimate the policy effects of companies using multiple languages to disclose ESG reports.

4. The authors must insert a discussions section. In the discussions section, the authors must be preset the results of the study in the context of similar papers. In this section, additional references can used (https://doi.org/10.1007/s43039-022-00056-x; https://doi.org/10.3390/su13168876)

Thanks for your helpful suggestion.

We refer to the references provided by reviewer 2, and we supplement the discussions section in the section "7 Conclusions and Discussion" of the paper. In the revised manuscript, the supplementary discussions section is as follows:

7.2 Further Discussion 

This paper makes important implications to both theory and practice. Theoretically, our study contributes to previous literature in the following ways.

First, this research contributes to the existing literature on ESG information disclosure by exploring how companies’ disclosure of ESG reports in multiple languages will affect the enthusiasm of foreign investors for holding shares. The research proposes that companies using multiple languages to disclose ESG reports can effectively increase the enthusiasm of foreign capital holdings, which is specifically reflected in the increase in the scale of foreign capital holdings and the increase in the number of foreign shareholders. Our research provides a new perspective on ESG information disclosure by highlighting the role of ESG reporting in multiple languages. The impact of corporate ESG rating levels on financial management and corporate governance has been discussed in a large amount of literature. This article focuses on the impact of ESG reporting information disclosure methods on foreign corporate investors. Our research shows that analyst attention and accounting information comparability are poor, and companies that hire non-Big Four auditing firms to audit financial reports are more enthusiastic about foreign shareholding disclosures in multilingual ESG reports. A large amount of research literature shows that ESG ratings affect shareholder governance and investor decision-making [65]. Our findings extend important perspectives in previous literature. We research the impact of companies on the interests of external investors from the perspective of ESG information disclosure, and improve the decision-making usefulness of Chinese companies’ ESG report disclosures in the international market.

Second, this research provides a new quasi-experimental scenario to explore the mechanism of the influence of ESG report disclosure language on cross-border capital flows, and helps to solve the endogenous causality of corporate governance behavior on ESG information disclosure in the existing literature. ESG information disclosure has become a broad consensus in the international capital market. The existing literature on corporate ESG metrics includes ESG ratings by ESG professional institutions, ESG performance scores and whether ESG report information is disclosed to explore the impact of corporate ESG performance on corporate decisions and behaviors [66-67]. Some scholars also try to reflect their alliances and cooperative actions from the common preferences of institutional investors for ESG, and clusters of ESG interests among institutional investors will work together to convince portfolio companies to innovate in low carbon [68]. However, the ESG performance of Chinese listed companies and the common ESG preferences of institutional investors are not an effective way to promote Chinese companies to convey ESG information to overseas investors. This paper identifies the information transmission methods that drive foreign investment into Chinese companies based on a quasi-natural experiment using a company to disclose ESG reporting information in multiple languages. Our research provides a useful roadmap for Chinese enterprises to transfer the performance of ESG responsibilities to the international capital market, which deserves more exploration in the following study.

Third, this research expands the existing literature on the functional effects of ESG information by exploring how companies use English to disclose ESG reports to improve the readability of corporate annual report information for foreign investors. Enterprises’ fulfillment of ESG responsibilities is an important way to promote high-quality economic development. However, the existing literature uses the evaluation indicators of ESG professional rating agencies to study the interaction between Chinese overseas investors and is very limited. This research find that the use of enterprises to disclose ESG reports in English has the enthusiasm of foreign investors to understand the information of corporate annual reports, which is more significant in the poor attention of analysts and the comparability of accounting information, and the enterprises that employ non-big four accounting firms to audit financial reports. This result is consistent with the popular view in the literature that annual report information readability affects capital market investor decisions [69-70]. This means that the information disclosure of the English version of ESG reports of Chinese listed companies is an effective means to reduce information asymmetry among foreign investors.

In fact, these findings have some implications for Chinese companies. Based on the above research conclusions, we draw the following three revelations. First, Chinese enterprises should strengthen the international versatility of information disclosure. Chinese companies should expand their attention from domestic investors to international investors based on the opening up of the Chinese market. Sustainable development goals are a topic of common concern to the international community. Chinese companies that use Chinese language to disclose operating reports only use the domestic capital market. To attract foreign investment, Chinese companies need to expand the international versatility of the company's report disclosure language.

Second, Chinese companies should strengthen international exchanges of ESG concepts. The sustainable development of Chinese enterprises is a topic of common concern in the international capital market. How foreign investors understand the ESG concepts and implementation of ESG responsibilities of Chinese enterprises is an important issue. This is also a key issue in promoting international capital circulation in China’s capital market. Chinese companies’ use of multiple languages to disclose ESG in Chinese and English will help strengthen the integration of corporate ESG information into the international capital market, promote Chinese foreign investors’ understanding and recognition of Chinese companies’ ESG information, and encourage foreign investors to be enthusiastic about holding shares in Chinese companies. 

Third, Chinese companies should consider the difference in effectiveness of the English version of the ESG report. If companies disclose their ESG reports in English, it will help enhance the readability of the company’s annual reports for foreign investors and alleviate the information asymmetry between foreign investors and companies. The positive effect of the English version of ESG reports on promoting foreign investors’ shareholdings may not always be satisfactory. As mentioned in this paper, the comparability of corporate accounting information and the supervision effect of external analysts and auditors can enhance the effect of the English version of the ESG report. The English version of ESG reports disclosed by Chinese companies makes up for the readability of annual report information and is an important mechanism to increase foreign shareholdings. Therefore, for Chinese enterprises to move toward the international capital market, providing readability of information disclosure is a perspective that promotes domestic and overseas capital circulation.

5.The authors must insert in the final section, considerations regarding limitations of the research and future directions for the research.

Thanks for your helpful suggestion.

 We refer to the references provided by reviewer 2, and we add research limitations and future directions in the last section of the paper. In the revised manuscript, the supplementary content is as follows:

8.Limitation and Future Research

This paper studies the impact of multilingual disclosure of ESG reports on the enthusiasm of foreign ownership. There are two limitations: firstly, this paper has limitations in the measurement of research variables. Within a fixed period, the degree of transaction volume completed in a short period of time can reflect the enthusiasm of foreign investment. Since it is impossible to observe the daily transaction time series of Chinese companies held by foreign investors, this paper uses the scale of domestic and foreign shareholdings in an accounting period and the number of foreign shareholders as proxy variables to measure the enthusiasm of foreign investors. Secondly, this paper has limitations in the research method. There are no unified standards for the disclosure content of corporate ESG reports, and there are no unified standards for the content of corporate ESG reports disclosed in multiple languages. This paper sets up dummy variables based on whether companies disclose ESG reports in Chinese and English, and uses a quasi-natural experiment to conduct empirical research as a preliminary exploration. After ESG reports are unified and standardized, the measurement method of ESG reports disclosed in multiple languages needs to be further improved.

There are more opportunities for future research on the topic of the impact of multilingual disclosure of ESG reports on the enthusiasm of foreign investors for holding shares. This paper puts forward research prospects in the following three aspects: firstly, the disclosure content of corporate ESG reports forms a unified normative standard. We can use text analysis to study the impact of the comparability of corporate ESG information disclosure in multiple languages on the behavioral decision-making of foreign investors. Secondly, China’s capital market is increasingly open to the world. After the information disclosure mechanism for foreign investors’ capital investment in China has been improved and matured, we further used survey research methods to study the impact of Chinese companies’ ESG information disclosure on the enthusiasm of foreign investors for holding shares. Thirdly, with the gradual improvement of corporate ESG report disclosure methods, the Chinese capital market has improved its mechanism for disclosing listed company information in multiple languages to the international market. The study of ESG report on the international capital market debt financing, import and export trade of enterprises needs to be explored.

Response to Reviewer 3

（PONE-D-23-30913）

1.I suggest to elaborate on the motivation of group variables choice and revise the hypotheses of the research to match the methodological process. Link the hypotheses with the discussions section of the paper and with the conclusions.

Thanks for your helpful suggestion.

Following the reviewer’s suggestion, we elaborated the motivation for the selection of group variables, clarified the research method, and modified the discussion part of the research hypothesis to make the research hypothesis, research method and research conclusion match.

In the revised manuscript, we have added the reasons for choosing the proxy variable of foreign shareholding enthusiasm as follows:

China vigorously promotes the construction of pilot free trade zones and free trade ports, actively promotes exchanges with international capital markets, and improves the degree of liberalization and facilitation for Chinese overseas investors, with a view to attracting a large amount of foreign investment into China’s capital market. The high-level opening up of China's capital market has further strengthened the confidence of global investors to participate and enhanced the enthusiasm of foreign investors for holding shares. China Securities Daily reported that there are two aspects that highlight the enthusiasm of foreign investors holding shares in Chinese listed companies. First, the scale of foreign investors purchasing shares in China's A-share market has increased. Second, the number of foreign investors has increased. We use the scale of foreign shareholding and the number of foreign shareholders to measure the enthusiasm of foreign investors to hold shares in Chinese listed companies.

In the revised manuscript, we have added the reasons for choosing control variables as follows:

Due to language and cultural barriers between foreign shareholders and local corporate managers and other shareholders, the information asymmetry between foreign shareholders and the company is serious, making it more difficult for foreign shareholders to grasp and predict the company’s future development and profitability. The future performance and income of a company affect the enthusiasm of foreign shareholders to invest in Chinese listed companies [10,28]. China has a vast territory, and differences in the level of market economic development in various regions affect the degree of attraction of the capital market to foreign investors[12]. The degree of diversification of corporate ownership structures affects the participation of major shareholders within the company in relation to foreign shareholders[10]. Enterprise ownership structure, financial performance characteristics and China’s regional macro factors affect the enthusiasm of foreign capital to invest in Chinese listed companies.

We revised the discussion of the research hypothesis and discussed that Chinese enterprises use multiple languages to disclose ESG reports, which affects the efficiency of information acquisition by overseas investors (Miller, 2010) and the accuracy of foreign investors' trust information (Lee, 2020). Based on this, we propose research hypothesis 1:.Multilingual ESG report disclosure is significantly positively related to the enthusiasm of foreign investors for holding shares. We have deleted research hypothesis 1-1 and research hypothesis 1-2 from the original manuscript.

In the revised manuscript, our revised research hypothesis is presented as follows:

Enterprises’ active performance of social responsibilities helps maintain and improve the relationship between enterprises and stakeholders. The stocks of enterprises with lower social responsibility performance can obtain excess returns in the short-term market, but corporate performance is unsustainable in the long-term market [44-45]. ESG reports carry non-financial information for companies to fulfill their social responsibilities. Text information is an explanation and supplement to digital information such as traditional financial statements, and has more information increments. ESG reports and financial reports have become the main basis for shareholders, creditors and other stakeholders to judge the sustainable development capabilities of enterprises. The ESG report contains a large amount of textual information. The textual content is forward-looking and predictive to a certain extent. Stakeholders can form expectations for the future development of the company based on the disclosed social responsibility information, which can help investors better understand the company's value and future prospects[46]. Currently, there are no international standards that uniformly regulate the disclosure content, form and language of corporate ESG reports. Companies can independently decide the content and form of ESG reports, which brings certain difficulties to investors in interpreting and analyzing ESG information. China's Shenzhen Stock Exchange and Shanghai Stock Exchange do not force listed companies to disclose relevant information reports in English. Most companies only disclose information reports in Chinese. If a company only discloses a Chinese version of its report, for foreign investors, communication language barriers will lead to a lack of readability of the ESG report. The readability of corporate annual reports directly affects the efficiency with which investors obtain information [47], and affects the accuracy of information that stakeholders trust [48].

First, the English version of ESG reports disclosed by companies affects the efficiency of foreign investors in obtaining information. Foreign investors’ participation in the Chinese market will help promote the effective flow and allocation of resources and increase stock liquidity [36]. The “Measures for the Administration of Information Disclosure by Listed Companies” promulgated by the China Securities Regulatory Commission stipulates that listed companies can publish materials in English and other languages on their own according to the needs of international development. As an annual report of corporate non-financial information, the ESG report discloses the English version of the ESG report. It is an international communication method and becomes a communication bridge between Chinese companies and foreign-owned companies. The ESG report is an annual report of corporate non-financial information. Disclosure of the English version of the ESG report is an international communication method and has become a bridge of communication between Chinese companies and foreign-owned shares. The English version of the ESG report can reduce the communication costs and information processing costs between foreign investors and Chinese enterprises, and solve the decision-making problems of overseas investors more conveniently and quickly. The English version of the ESG report improves the information environment of the capital market, broadens the efficiency of information use in the international market, helps improve the decision-making quality of information users in the capital market, and increases the willingness of foreign investors to trade [2].

Second, the English version of ESG reports disclosed by companies affects the accuracy of the information that foreign investors trust. Corporate social responsibility is a kind of “on-the-job consumption” of management. Management often over-invests in social responsibility-building activities to enhance their professional reputation and personal reputation. Social responsibility disclosure may further strengthen management's tendency to violate regulations [49]. CSR disclosures provide management with a “legitimate justification” and a “license” to engage in financial irregularities [50]. Therefore, the communication costs between foreign investors and local corporate managers and other shareholders are usually higher, making the information asymmetry between foreign investors and insiders more serious. Due to the profit-seeking nature of capital, the future performance and earnings of enterprises are the focus of investors' decision-making. Compared with local Chinese investors, it is more difficult for foreign investors to grasp and predict the company's future development and profitability. Companies disclosing ESG reports in English can help enhance the readability of corporate ESG report information for foreign investors. The easier it is for foreign investors to interpret the contents of ESG reports of Chinese companies, the more accurately they can judge the company's operating conditions and investment risks [41]. Choosing English to disclose information in ESG reports maximizes the familiarity of foreign shareholders, enhances the impact of corporate internationalization, and promotes foreign shareholders’ recognition of corporate culture and identity. Foreign investors have reduced their doubts about the operation and management of Chinese companies, increased their trust in companies’ public disclosure of information, and increased investors’ willingness to invest [51].

Based on the research hypothesis and research variables, we explain the reasons for choosing the DID research method in this paper and the research hypothesis tested by the DID research method.

2.Does the DID estimator appear in any equation? If yes, a proper identification and description should be provided. In the current form of the paper, the theoretical discussions of the results do not seem to be clearly linked to the description of equations.

Thanks for your helpful suggestion.

We detail the variables and estimators of the DID model (1) to (7). We explain the variables and estimators in detail below each DID model. In the revised manuscript, we add the following:

 (1)

 (2)

In the difference-in-differences model (1) and (2), Time is a dummy variable. Before the English version of the ESG report is disclosed, the variable Time is assigned a value of 0. After the English version of the ESG report is disclosed, the variable Time is assigned a value of 1. Before 2018, the sustainable development information of listed companies in China regarding the fulfillment of social responsibilities and other the disclosure is voluntary. Articles 95 and 96 of China’s 2018 “Code of Governance for Listed Companies” mandate that companies disclose environmental information, fulfill social responsibilities and corporate governance. Starting in 2018, it is officially confirmed that corporate ESG reports can be disclosed in multiple languages. The experimental group represents companies that disclose ESG reports in both Chinese and English, and is assigned Treat=1. The control group represents companies that only disclose ESG reports in Chinese, and is assigned Treat=0. Size, Age, Roa, Lev, Lon, Top1, Soe, LnGDP, City_employ, and City_R&D represent the control variables. γi ,ηi ,λt andμi represent the fixed effect of the control area, the fixed effect of the technology-intensive industry, the time fixed effect, and the individual enterprise effect, respectively. εi,t is the random error term of the model, in order to further avoid the impact of unobservable variables on the empirical results of the paper. The variable we are interested in is the interaction term Time×Treat, which represents that companies’ use of multiple languages to disclose ESG reports increases the enthusiasm of foreign investors. We expect β1 to be positive. The β2~β11 represent the regression coefficients corresponding to the control variables.

 (3)

 (4)

In the difference-in-differences model (3) and (4), the variables Befi,s, Curi,t and Afti,m respectively represent the cross-multiplication term of the year dummy variable and the corresponding experimental group dummy variable s years before the policy implementation, the t year of the current policy implementation period, and m years after the policy implementation. β1, β2, and β3 represent the regression coefficients of variables Befi,s, Curi,t, and Afti,m respectively. Size, Age, Roa, Lev, Lon, Top1, Soe, LnGDP, City_employ, and City_R&D represent the control variables. The β4~β13 represent the regression coefficients corresponding to the control variables. γi ,ηi ,λt and μi represent the fixed effect of the control area, the fixed effect of the technology-intensive industry, the time fixed effect, and the individual enterprise effect, respectively. εi,t is the random error term of the model.

 (5)

 (6)

 (7)

Among them, Time is a dummy variable. Starting in 2018, it is officially confirmed that corporate ESG reports can be disclosed in multiple languages. Before the English version of the ESG report is disclosed, the variable Time is assigned a value of 0. After the English version of the ESG report is disclosed, the variable Time is assigned a value of 1. The experimental group represents companies that disclose ESG reports in both Chinese and English, and variable Treat is assigned a value of 1. The control group represents companies that only disclose ESG reports in Chinese, and variable Treat is assigned a value of 0. We expect that corporate disclosure of ESG reports in multiple languages will help improve the readability of the company’s annual report, and the regression coefficient β1 of the difference-in-difference model (5) will be negative. The variable Readability is added to the difference-in-differences model (6) and (7) for testing. We predict that the readability difficulty of the company’s annual report will be significantly negatively related to the enthusiasm of foreign investors, and the regression coefficient β1 will be negative.

Size, Age, Roa, Lev, Lon, Top1, Soe, LnGDP, City_employ, and City_R&D represent the control variables. In the difference-in-difference model (5), β2~β11 represent the regression coefficients corresponding to the control variables. In the difference-in-differences model (6) and (7), β2 represents the regression coefficient of the mediating variable Readability, and β3~β11 represent the regression coefficients corresponding to the control variables. γi ,ηi ,λt andμi represent the fixed effect of the control area, the fixed effect of the technology-intensive industry, the time fixed effect, and the individual enterprise effect, respectively. εi,t is the random error term of the model.

3.The explanations to all equations should be revised or completed so that explanations are provided for all used symbols. For example, equations (3) and (4) use regression coefficients β1 to β17, but the explanations in lines 453-456 refer to coefficients β1 to β3 and, additionally, refer to dummy variables S years and M years which are not present in the corresponding equations. Equation (11) needs additional explanations like why subtraction operation is used for different companies accounting systems and what t=15 and 1/16 stand for. In line 642, coefficient is mentioned with reference to model (8), but model (8) does not include such coefficient. Please check and correct.

Thanks for your helpful suggestion.

We provide explanations of the symbols used in all equations. We detail the meaning of the variables and coefficients in each equation.

 (3)

 (4)

In the difference-in-differences model (3) and (4), the variables Befi,s, Curi,t and Afti,m respectively represent the cross-multiplication term of the year dummy variable and the corresponding experimental group dummy variable s years before the policy implementation, the t year of the current policy implementation period, and m years after the policy implementation. β1, β2, and β3 represent the regression coefficients of variables Befi,s, Curi,t, and Afti,m respectively. Size, Age, Roa, Lev, Lon, Top1, Soe, LnGDP, City_employ, and City_R&D represent the control variables. The β4~β13 represent the regression coefficients corresponding to the control variables. γi ,ηi ,λt and μi represent the fixed effect of the control area, the fixed effect of the technology-intensive industry, the time fixed effect, and the individual enterprise effect, respectively. εi,t is the random error term of the model.

We check the proxy indicator data of the comparability of accounting information and the expression of the formula in the paper. We found that the accounting information comparability of proxy indicator data is accurate. In the process of writing the original manuscript, we did not clearly explain the construction process of proxy indicators to express the comparability of accounting information. We are very sorry that we have caused reading difficulties to the reviewers.

We have revised the expression of accounting information comparability. The coefficients mentioned in the original manuscript model (8) have been added to our revised version. The subtraction operation used in the accounting system represents the difference between the expected earnings of company i and the expected earnings of company j, which indicates the comparability of the accounting information of the two companies. The number 1/16 of equation (11) in the original manuscript represents the average of the difference in expected earnings between the two companies for the previous 16 quarters, which makes the proxy indicator of comparability of accounting information more robust.

In the original manuscript equation (11), t=15, we corrected it to t=16 in the revised manuscript, which represents the absolute sum of the difference between the expected accounting earnings of the two companies in 16 consecutive quarters.

4.Explain the meaning of numbers in parentheses in Tables 4, 5, 6, 7, 8, 9, 10 preferably at the bottom of each table.

Thanks for your helpful suggestion.

According to the suggestions of reviewer 1, reviewer 2 and reviewer 3, our revised paper has 12 tables. Below Tables 4, 5, 6, 7, 8, 9, 10, 11, and 12 we annotate the meaning of the numbers in brackets in the tables.

5. Please check the meaning of the following phrases and revise them:

- lines 32-34 (”Financial performance reporting, which will help foreign investors better understand the company’s operations.”),

- lines 539-540 (”They provide investors with incremental information by asking questions and observing management’s tone and tone.”),

- lines 596-597 (”This paper uses whether the big four international accounting firms are hired for corporate financial report auditing to measure the audit status of the big four international auditing firms.”)

Thanks for your helpful suggestion.

According to the reviewer's comments, we modify as follows:

About “Financial performance reporting, which will help foreign investors better understand the company’s operations.”,we amend it as “The Spanish investor suggested that companies listed on the Shanghai Stock Exchange and Shenzhen Stock Exchange publish financial performance reports in English like companies listed in Europe and Hong Kong, which would help foreign investors better understand the company’s operations.”

About “They provide investors with incremental information by asking questions and observing management's tone and tone.”, we amend it as “Analysts provide incremental information to investors by observing the tone and intonation of management's. responses to questions.”

About “This paper uses whether the big four international accounting firms are hired for corporate financial report auditing to measure the audit status of the big four international auditing firms.”, we amend it as “If an enterprise hires a big four international auditing firm to audit the financial report, we define the value as 1 for the Big Four; otherwise, the value is 0.”

6.Explain the meaning of used abbreviation the first time you use them: PCAOB in line 194, CSMAR, CNRDS in line 310, ST and ST* in line 312, PSM in line 315. Also, explain the meaning of the word ”tertiles” used in several parts of the paper.

Thanks for your helpful suggestion.

In the paragraph where the PCAOB first appeared, we clearly marked that the full name of the PCAOB is “the Public Company Accounting Oversight Board.” CSMAR, CNRDS, ST and ST*, PSM, we footnote the full names. CSMAR is the abbreviation of China Stock Market & Accounting Research Database. CNRDS is the abbreviation of Chinese Research Data Services Database. ST indicates that the company's net income has been negative in the past two years, that is, the listed company has suffered losses for two consecutive years or its net assets are lower than the face value of the stock. Or the company’s shareholders’ rights in the last year were lower than its registered capital. *ST means that the company has suffered losses for three consecutive years and is at risk of being withdrawn from the stock market. PSM is the abbreviation of propensity score matching method.

We revise every paragraph where “tertiles” appeared. First, we rank companies from smallest to largest by industry and year based on the moderating variable indicator. Second, we divided the sorted sample into three groups based on the thirds and thirds. Third, we performed group regression tests using the top third group sample and the bottom third group sample.

7.Avoid using capital letters for ”This paper” within the phrase and correct lines 306 and 308.

Thanks for your helpful suggestion.

We have corrected the use of capital letters "This paper" in the phrase in lines 306 and 308.

8.According to subsection 4.1., data was collected from CSMAR database and CNRDS database, but the Data availability statement mentions that the used data are owned by CSMAR and CICPA. Please explain.

Thanks for your helpful suggestion.

All data used in the study are publicly available from sources cited in the text. Interested persons can contact CSMAR for the data (see https://www.gtarsc.com/ for more details, contact via 400-639-888), CNRDS(see https://www.cnrds.com/for more details, contact via 021-66181082), Official website of Shenzhen Stock Exchange(see https://www.szse.cn/for more details, contact via 400-808-9999) and Official website of Shanghai Stock Exchange(see https://www.ssse.cn/for more details, contact via 400-8888-400). We confirm that we do not have any special access or privileges to the data that other researchers would not have.

The other two revisions of this paper are as follows:

First, we revised the discussion in section 6.1 Inspection of impact mechanism to make the content of this section match the research hypothesis and conclusion. In the revised draft, our specific modifications are as follows:

Most Chinese domestic enterprises disclose ESG reports in Chinese for domestic capital market investors. To achieve the goal of world-class enterprises, Chinese enterprises need ESG concepts to be transmitted to the international capital market. The ESG information of Chinese enterprises is internationalized, and the use of English to disclose ESG reports can help enhance the efficiency of foreign investors in obtaining ESG information of Chinese enterprises, and strengthen foreign investors’ trust in the accuracy of ESG information of Chinese enterprises. Investment decision needs the reliability of financial information. The disclosure of ESG report is an important non-financial information report of an enterprise. The environmental responsibility, social responsibility and corporate governance concerned in ESG report can all be reflected in the financial information of an enterprise. ESG report disclosure and financial information become complementary functions, which can become the information basis for investors to understand the authenticity of financial information. Chinese enterprises’ use of English to disclose ESG reports has improved the readability of corporate financial reports for foreign investors.

Chinese enterprises use English to disclose ESG reports to supplement the difficulty of foreign investors in interpreting Chinese enterprises’ financial reports and enhance the readability of foreign investors’ financial reports. The improvement of the readability of the text helps to enhance the efficiency of foreign investors in obtaining information. By using the ESG reporting information of Chinese enterprises, foreign investors can understand the financial reporting information of Chinese enterprises, understand the implementation of ESG responsibilities of enterprises, and further trust the authenticity of financial information. Low readability of annual report text information will reduce investors’ investment in enterprises [51]. Multilingual disclosure of ESG reports by enterprises can promote the readability and understanding of financial information by foreign investors, which is an influence mechanism to stimulate the enthusiasm of foreign shareholding.

Second, according to the reviewer’s suggestion, we added the test of the moderating effect of regional system and regional culture. In the conclusion part of the study, we add the conclusion of the moderating effect of regional institutions and regional culture.

---

## [Decision Letter · Decision Letter 1]

22 Jan 2024

PONE-D-23-30913R1The impact of companies disclosing ESG reports in multiple languages on the enthusiasm of foreign investors for holding sharesPLOS ONE

Dear Dr. Bao,

Thank you for submitting your manuscript to PLOS ONE. After careful consideration, we feel that it has merit but does not fully meet PLOS ONE’s publication criteria as it currently stands. Therefore, we invite you to submit a revised version of the manuscript that addresses the points raised during the review process.

We look forward to receiving your revised manuscript.

Kind regards,

Ionela Munteanu, PhD

Academic Editor

PLOS ONE

Journal Requirements:

Additional Editor Comments (if provided):

The text needs additional revision related to typographical errors and apparent inaccuracies in the references. Please verify whether the numbers mentioned in the text should be cross-referenced with entries in your reference list.

Reviewers' comments:

Reviewer's Responses to Questions

**Comments to the Author**

1. If the authors have adequately addressed your comments raised in a previous round of review and you feel that this manuscript is now acceptable for publication, you may indicate that here to bypass the “Comments to the Author” section, enter your conflict of interest statement in the “Confidential to Editor” section, and submit your "Accept" recommendation.

Reviewer #2: All comments have been addressed

Reviewer #3: All comments have been addressed

2. Is the manuscript technically sound, and do the data support the conclusions?

Reviewer #2: Yes

Reviewer #3: Yes

3. Has the statistical analysis been performed appropriately and rigorously? 

Reviewer #2: Yes

Reviewer #3: Yes

4. Have the authors made all data underlying the findings in their manuscript fully available?

Reviewer #2: No

Reviewer #3: Yes

5. Is the manuscript presented in an intelligible fashion and written in standard English?

Reviewer #2: Yes

Reviewer #3: No

6. Review Comments to the Author

Reviewer #2: The authors substantially improved the manuscript according to the recommendations made by the reviewers. I consider that the manuscript can be published

Reviewer #3: Dear authors,

Thank you for your efforts in improving the manuscript. I consider that all the recommendations have been addressed in an acceptable manner.

There are still some corrections to be made within the text concerning minor typos and apparently errors for references. Please check if the numbers that appear within the text should be linked to references in your reference list.

Good luck with your future research!

7. PLOS authors have the option to publish the peer review history of their article (what does this mean?). If published, this will include your full peer review and any attached files.

Reviewer #2: No

Reviewer #3: No

---

## [Author Response · Author response to Decision Letter 1]

26 Jan 2024

Dear editor,

Our most sincere thanks to you, Reviewer 3 for the very thoughtful review and helpful comments, and suggestions on our writing in this round.

We provide the original comments from Reviewer 3 in boldface and our responses to each item in regular font. We highlight the main amendment in blue in the revised manuscript.

Response to Reviewer 3

（PONE-D-23-30913R1）

1.There are still some corrections to be made within the text concerning minor typos and apparently errors for references. 

Thanks for your helpful suggestion.

According to the reviewer’s comments, we have revised the manuscript’s format, punctuation and typos. There are some grammatical errors in the sentences, and we have corrected them. In the manuscript, we have highlighted the revisions in the text in blue font.

The references corrected in the manuscript are highlighted in blue. The collated references are listed as follows:

1.Eleonora,B.Capital Structure and Innovation:Causality and Determinants.Empirical .2013;40(1),111-151.https://doi.org/10.1007/s10663-011-9179-y.

2.Hain, D., Johan, S., Wang, D. Determinants of Cross-Border Venture Capital Investments in Emerging and Developed Economies: The Effects of Relational and Institutional Trust. Journal of Business Ethics.2016;138:743–764. https://doi.org/10.1007/s10551-015-2772-4. 

3.Chen H,Chen J Z,Lobo G J,et al.Effect s of audit quality on earnings management and cost of equity capital:Evidence from China.Contemporary Accounting Research.2011; 28(3):892-925.https://doi.org/10.1111/j.1911-3846.2011.01088.x.

4.An, J., Jiang, M., Xu, J.Professional norms and risk-taking of bank employees: Do expectations of peers’ risk preferences matter?. Journal of Financial Stability.2021; 56(C):1-13.https://doi.org/10.1016/j.jfs.2021.100938.

5.So Ra Park, Jae Young Jang.The lmpact of ESG Management on Investment Decision: InstitutionalInvestors’ Perceptions of Country-Specific ESG Criteria.International Journal Financial Studies. 2021; 9(3): 1-27.https://doi.org/10.3390/ijfs9030048

6.llze Zumente, Natalja Lace.ESG Rating Necessity for the Investor or the Company?.Sustainability. 2021;13(16):1-14. https://doi.org/10.3390/su13168940.

7.Bao Wu,Qiuyang Gu,Zijia Liu,Jiaqiang Liu.Clustered Institutional Investors, Shared ESG Preferences, and Low-carbon Innovation in Family Firms.Technological Forecasting and Social Change.2023;194(1):1-25.https://doi.org/10.1016/j.techfore.2023.122676.

8.Hatem Riiba, Samir Saadi, Sabri Boubaker, Sara Xiaova Ding.Annual report readability andthe cost of equity capital.Journal of Corporate Finance.2021;67(C):1-25.https://doi.org/101902.10.1016/j.jcorpfin.2021.101902.

2.Please check if the numbers that appear within the text should be linked to references in your reference list.

Thanks for your helpful suggestion.

After carefully reading the reviewers’ comments and manuscripts, we adopted footnotes to better explain the source of the data in the text.

As a footnote, we explain the numbers mentioned within the text, namely that “less than 2% of listed Chinese companies have ESG reporting checks” and “China has become the world’s largest FDI outflow, with a total investment of $133 billion.”

The numbers mentioned in “less than 2% of listed companies in China have conducted ESG report verification” is our manual statistical collation based on ESG verification reports issued by China’s Shenzhen Stock Exchange and Shanghai Stock Exchange from 2006 to 2022. Our statistical results are listed in the appendix.

“China has become the world's largest FDI outflow with a total investment of US $133 billion”, according to the World Investment Report 2021, available on the UNCTAD website (https://unctad.org/).

Appendix

Statistics on disclosure and assurance of ESG reports, 2006-2022

Year Number of listed enterprises The number of ESG reports disclosed by companies Percentage of companies that disclose ESG reports Number of ESG confirmations The proportion of ESG report assurance to the number of ESG reports disclosed by enterprises

2006 1434 1340 93.44% 0 0.00%

2007 1550 1460 94.19% 0 0.00%

2008 1625 1513 93.11% 11 0.73%

2009 1718 1657 96.45% 10 0.60%

2010 2063 2002 97.04% 19 0.95%

2011 2342 2227 95.09% 22 0.99%

2012 2472 2345 94.86% 38 1.62%

2013 2489 2390 96.02% 34 1.41%

2014 2587 2500 96.64% 33 1.31%

2015 2806 2688 95.79% 30 1.11%

2016 3052 2977 97.54% 38 1.26%

2017 3440 3353 97.47% 55 1.62%

2018 3567 3454 96.83% 48 1.37%

2019 3777 3659 96.88% 44 1.18%

2020 4140 4106 99.18% 82 1.94%

2021 4669 4628 99.12% 127 2.62%

2022 5157 1732 33.59% 48 2.77%

The disclosure rate of listed companies’ ESG reports also reached more than 90%. The rate of ESG reports assurance is less than 2% by 2020, and the rate of ESG reports assurance is 2% or more by 2021-2022.

---

## [Editor Report · Decision Letter 2]

7 Feb 2024

PONE-D-23-30913R2The impact of companies disclosing ESG reports in multiple languages on the enthusiasm of foreign investors for holding sharesPLOS ONE

Dear Dr. Bao,

Thank you for submitting your manuscript to PLOS ONE. After careful consideration, we feel that it has merit but does not fully meet PLOS ONE’s publication criteria as it currently stands. Therefore, we invite you to submit a revised version of the manuscript that addresses the points raised during the review process.

 Specifically: Thank you for your effort in improving your paper. 

Please check the numbers mentioned within the text that should correspond to references in your reference list and put them in brackets.For example, the last paragraph before the Literature review reads: "One type of literature is the study of the effects of the introduction of different types of equity on corporate governance and corporate performance 10. Another type of literature is the study of factors affecting changes in corporate equity size 11. Research on the introduction of cross-border capital by enterprises focuses on the effectiveness of equity governance effects 12, and few literatures focus on the advantageous strategies of Chinese enterprises in attracting cross-border capital 13." --- This paragraph contains the numbers 10, 11, 12, 13. Does these numbers correspond to citations in your reference list? If so, please put the numbers in brackets. Proceed so with all similar cases.In the above mentioned paragraph, please replace "few literatures" with "few scholars" or "few manuscripts".Additionally, ensure that all references listed appear as citations within the text. For example, I could not identify the in-text citation of the reference numbered [70] in your reference list.

We look forward to receiving your revised manuscript.

Kind regards,

Ionela Munteanu, PhD

Academic Editor

PLOS ONE
---

## [Author Response · Author response to Decision Letter 2]

9 Feb 2024

ost sincere thanks to you, editor for the very thoughtful review and helpful comments, and suggestions on our writing in this round.

We provide the original comments from editor in boldface and our responses to each item in regular font. We highlight the main amendment in blue in the revised manuscript.

Response to Editor

（PONE-D-23-30913R2）

1.For example, the last paragraph before the Literature review reads: “One type of literature is the study of the effects of the introduction of different types of equity on corporate governance and corporate performance 10. Another type of literature is the study of factors affecting changes in corporate equity size 11. Research on the introduction of cross-border capital by enterprises focuses on the effectiveness of equity governance effects 12, and few literatures focus on the advantageous strategies of Chinese enterprises in attracting cross-border capital 13.” --This paragraph contains the numbers 10, 11, 12, 13. Does these numbers correspond to citations in your reference list? If so, please put the numbers in brackets. Proceed so with all similar cases.

Thanks for your helpful suggestion.

We have reviewed the full text. In the text, we put the citation numbers for the references in brackets.

2.In the above mentioned paragraph, please replace “few literatures” with “few scholars” or “few manuscripts”.

Thanks for your helpful suggestion.

We replace “few literatures” with “few scholars”.

3.Additionally, ensure that all references listed appear as citations within the text. For example, I could not identify the in-text citation of the reference numbered [70] in your reference list.

Thanks for your helpful suggestion.

In the text, “the reference numbered [70]” is in the fourth paragraph of “7.2 Further Discussion”.

---

## [Editor Report · Decision Letter 3]

13 Feb 2024

The impact of companies disclosing ESG reports in multiple languages on the enthusiasm of foreign investors for holding shares

PONE-D-23-30913R3

Dear Dr. Bao,

We’re pleased to inform you that your manuscript has been judged scientifically suitable for publication and will be formally accepted for publication once it meets all outstanding technical requirements.

Kind regards,

Ionela Munteanu, PhD

Academic Editor

PLOS ONE

---

## [Editor Report · Acceptance letter]

27 Feb 2024

PONE-D-23-30913R3 

PLOS ONE

Dear Dr. Bao, 

I'm pleased to inform you that your manuscript has been deemed suitable for publication in PLOS ONE. Congratulations! Your manuscript is now being handed over to our production team.

Kind regards, 

on behalf of

Dr. Ionela Munteanu 

Academic Editor

PLOS ONE